# Phylogenomic Analyses of *Snodgrassella* Isolates from Honeybees and Bumblebees Reveal Taxonomic and Functional Diversity

Luc Cornet,[a] Ilse Cleenwerck,[b] Jessy Praet,[b] Raphaël R. Leonard,[c] Nicolas J. Vereecken,[d] Denis Michez,[e] Guy Smagghe,[f]
[ID] Denis Baurain,[c] [ID] Peter Vandamme[b]

[a]BCCM/IHEM, Mycology and Aerobiology, Sciensano, Brussels, Belgium
[b]Laboratory of Microbiology and BCCM/LMG Bacteria Collection, Faculty of Sciences, Ghent University, Ghent, Belgium
[c]InBioS–PhytoSYSTEMS, Eukaryotic Phylogenomics, University of Liège, Liège, Belgium
[d]Agroecology Lab, Université libre de Bruxelles (ULB), Brussels, Belgium
[e]Laboratory of Zoology, Research Institute for Biosciences, University of Mons, Mons, Belgium
[f]Laboratory of Agrozoology, Faculty of Bioscience Engineering, Ghent University, Ghent, Belgium

**ABSTRACT** *Snodgrassella* is a genus of *Betaproteobacteria* that lives in the gut of honeybees (*Apis* spp.) and bumblebees (*Bombus* spp). It is part of a conserved microbiome that is composed of a few core phylotypes and is essential for bee health and metabolism. Phylogenomic analyses using whole-genome sequences of 75 *Snodgrassella* strains from 4 species of honeybees and 14 species of bumblebees showed that these strains formed a monophyletic lineage within the *Neisseriaceae* family, that *Snodgrassella* isolates from Asian honeybees diverged early from the other species in their evolution, that isolates from honeybees and bumblebees were well separated, and that this genus consists of at least seven species. We propose to formally name two new *Snodgrassella* species that were isolated from bumblebees: i.e., *Snodgrassella gandavensis* sp. nov. and *Snodgrassella communis* sp. nov. Possible evolutionary scenarios for 107 species- or group-specific genes revealed very limited evidence for horizontal gene transfer. Functional analyses revealed the importance of small proteins, defense mechanisms, amino acid transport and metabolism, inorganic ion transport and metabolism and carbohydrate transport and metabolism among these 107 specific genes.

**IMPORTANCE** The microbiome of honeybees (*Apis* spp.) and bumblebees (*Bombus* spp.) is highly conserved and represented by few phylotypes. This simplicity in taxon composition makes the bee's microbiome an emergent model organism for the study of gut microbial communities. Since the description of the *Snodgrassella* genus, which was isolated from the gut of honeybees and bumblebees in 2013, a single species (i.e., *Snodgrassella alvi*), has been named. Here, we demonstrate that this genus is actually composed of at least seven species, two of which (*Snodgrassella gandavensis* sp. nov. and *Snodgrassella communis* sp. nov.) are formally described and named in the present publication. We also report the presence of 107 genes specific to *Snodgrassella* species, showing notably the importance of small proteins and defense mechanisms in this genus.

**KEYWORDS** honeybee, bumblebee, microbiome, metagenomics, *Snodgrassella*, phylogenomics, species delineation, functional analysis, metabolic modeling

Address correspondence to Peter Vandamme, peter.vandamme@ugent.be.

The authors declare no conflict of interest.

Honeybees (*Apis* spp.) and bumblebees (*Bombus* spp.) harbor a gut microbiome that is important in health and metabolism (1–3). This microbiome is highly conserved, with 95% of the gut microbionts falling within a few phylotypes that include *Actinobacteria* (*Bifidobacterium*, *Bombiscardovia*), *Bacteroidetes* (*Apibacter*), *Firmicutes* (*Lactobacillus* [the so-called Firm-5 or Lacto-1 taxon], *Bombilactobacillus* [Firm-4, Lacto-2], and *Apilactobacillus*

[Lacto-3]), *Alphaproteobacteria* (*Bartonella*, *Bombella*, *Commensalibacter*), *Betaproteobacteria* (*Snodgrassella*), and *Gammaproteobacteria* (*Frischella*, *Gilliamella*) (2, 4–18). The genera *Bifidobacterium*, *Lactobacillus*, *Bombilactobacillus*, *Gilliamella*, and *Snodgrassella* are generally considered the core microbionts of honeybees and bumblebees (10, 19, 20). These gut-related organisms coevolved within their hosts during the last 80 million years (9, 10, 21) and contribute to carbohydrate digestion (2, 18, 22) and pathogen defense (23–26). The importance of bees for ecosystem integrity, the contribution of the bee gut microbiota to its hosts' health, and the relative simplicity of the taxonomic composition of the bee gut microbiota, along with their mode of transmission, which is mainly vertical, have made bees an emerging model organism for the study of gut-related microbial communities (1, 18, 27–29).

*Snodgrassella* and *Gilliamella* are physically closely associated within the hindgut: the former grows in contact with the ileum epithelium, while the latter forms a dense biofilm on top of the *Snodgrassella* layer and is in contact with the gut lumen (9, 11, 30). As a result of intricate coevolution of these two bacteria, horizontal gene transfers (HGTs) have been reported between these two taxa (2, 31). Among others, they share Rhs toxin proteins, which are involved in type VI secretion system (T6SS)-mediated competition (2, 32–34).

The formal description and naming of *Snodgrassella alvi* were based on three strains: one from a honeybee gut sample (wkB2$^T$ from *Apis mellifera*) and two from bumblebee gut samples (wkB12 from *Bombus bimaculatus* and wkB29 from *Bombus vagans*) (21). All three strains shared more than 99.1% of their small subunit (SSU [16S]) rRNA gene sequences, which allocated them within the *Neisseriaceae* family. Many additional strains have been reported since, and some phylogenetic analyses revealed a clear separation between *Snodgrassella* isolates of *Apis* spp. and those of *Bombus* spp. (1, 32–35), whereas others reported that the two groups were not monophyletic (10, 22). In the present study, we used core gene phylogenomics to demonstrate the separation of *Snodgrassella* isolates into three clades: one colonizing *Bombus* spp. and two colonizing *Apis* spp. We report an early divergence of *Snodgrassella* strains from Asian honeybees, which implies that *Apis*-colonizing strains are paraphyletic. We used average nucleotide identity (ANI) analyses to demonstrate that the genus *Snodgrassella* consists of at least seven species. We further analyzed differential gene contents within these species and examined evolutionary scenarios for the emergence of 107 specific genes. We finally used our own *Snodgrassella* isolates to describe and formally name two of these novel species from bumblebees as *Snodgrassella gandavensis*, with LMG 30236 (=CECT 30450) as the type strain, and *Snodgrassella communis*, with LMG 28360 (=CECT 30451) as the type strain.

## RESULTS AND DISCUSSION

**Species delimitation within the *Snodgrassella* genus.** We studied the diversity within the genus *Snodgrassella* using extensive phylogenomic and average nucleotide identity (ANI) analyses (Fig. 1). Highly conserved genes were selected to perform the phylogenomic analyses. Multiple events of HGT affecting *Snodgrassella* genomes have been reported (2, 31, 32), and such events may be damaging for inferring species phylogenies (36–38). To minimize interference of HGT, we used shared *Neisseriaceae* core genes only, by incorporating 35 non-*Snodgrassella* bacteria from this family into our data set. We further imposed a strict unicopy presence onto core genes in order to avoid artifacts linked to paralogous sequences, which also have been reported to be deleterious to phylogenomic studies (39, 40). Finally, we enforced high geometric and functional indices (0.8 for both), resulting in the selection of genes with less than 20% gap insertions and 20% substitutions, respectively. Our phylogenomic data set was thus composed of 254 genes that were structurally and functionally conserved within the family *Neisseriaceae*. Our analyses of the *Neisseriaceae* phylogeny confirmed with 100% bootstrap support that the genus *Snodgrassella* is monophyletic within the family *Neisseriaceae* (see Fig. S1 in the supplemental material) (21, 41). Its nearest-neighbor taxon was *Populibacter corticis* (40). Prior to the description of *P. corticis* in 2017 (41), the lineage formed by *Stenoxybacter acetivorans* and *Snodgrassella* spp. was suggested to represent a gut-specific clade within the *Betaproteobacteria*, based on the observation that also *S. acetivorans* was isolated from an insect (i.e., termite gut sample) (21). The description of *P. corticis*, an organism isolated from bark tissue of poplar canker, appeared to disrupt this image, yet, more

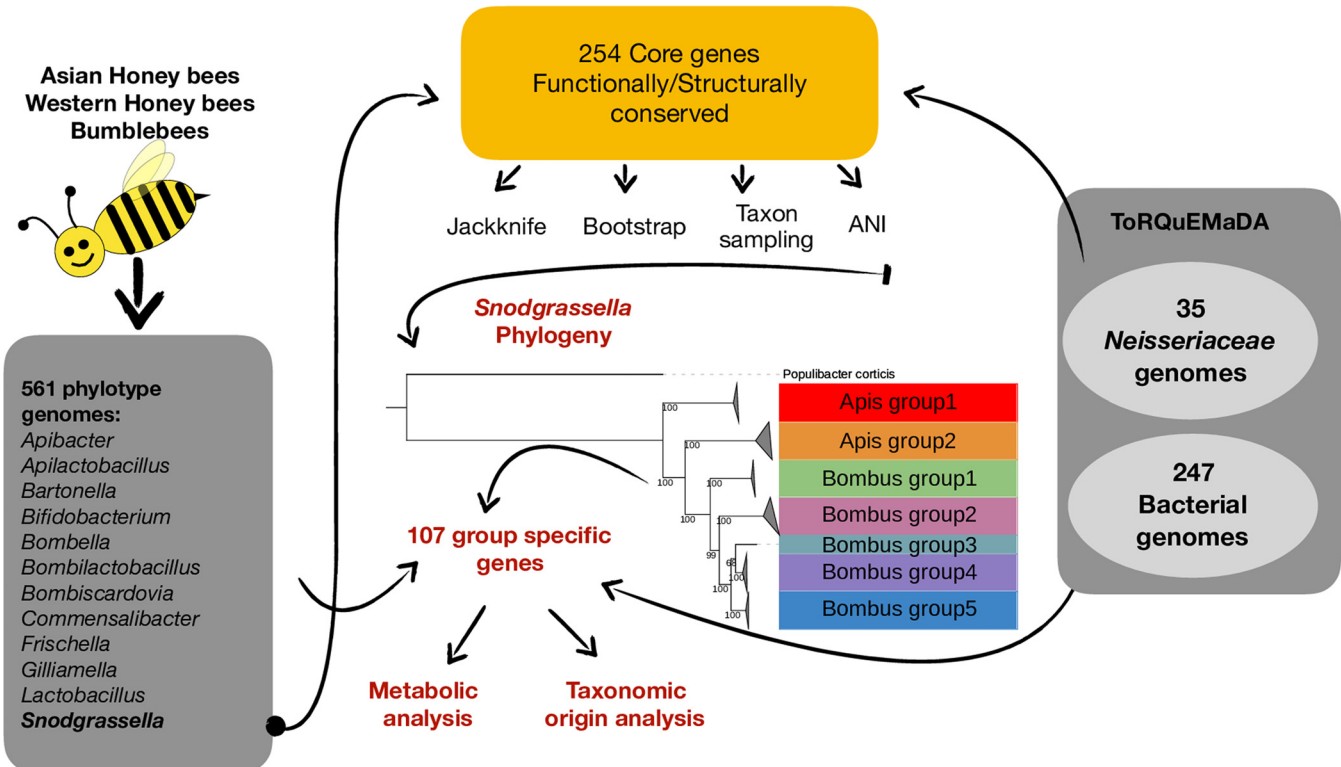

**FIG 1** Graphical abstract. We used 9 newly sequenced *Snodgrassella* genomes and 67 (including the outgroup) public genome assemblies to reconstruct the phylogeny of the genus *Snodgrassella* with 254 core genes, all functionally and structurally conserved in the *Neisseriaceae* family. We used independent phylogenetic methods (bootstrapping, leave-one-out analysis, and taxon sampling) to assess the robustness of our phylogenomic trees. Combined with an average nucleotide identity analysis, our results indicated new species delimitations within this genus. We further investigated specific gene contents within the species groups and their functional importance.

should be known about the latter organism before the hypothesis of a gut-specific clade within the *Betaproteobacteria* is abandoned. In the rest of the present study, we used *P. corticis* CFCC 13594[T] as an outgroup for studying *Snodgrassella* phylogeny.

Our amino acid- and nucleotide-based phylogenomic and leave-one-out analyses all revealed a clear separation between *Snodgrassella* isolates from *A. mellifera* and *Bombus* spp., with full support values in bootstrap and leave-one-out analyses (Fig. S2 and S3). The 75 *Snodgrassella* strains formed three clades. Those isolated from Asian honeybees, referred to as Apis group1, formed a monophyletic group separated from the two other clades with full support in both bootstrap and leave-one-out analyses (Fig. S2 and S3). The same three clades were reported by Powell et al. (42) using *minD* as a single phylogenetic marker gene. Yet, the latter study did not use an outgroup to root the obtained phylogeny. Using 254 core genes and *P. corticis* CFCC 13594[T] as an outgroup, we demonstrated that Apis group1 is the first diverging group of the *Snodgrassella* phylogeny (Fig. S2 and S3). The separation into three lineages was confirmed by removing *P. corticis* CFCC 13594[T] from the phylogenomic analyses in order to check the presence of a long-branch attraction artifact (Fig. S4). The *Snodgrassella* phylogeny showed shorter branch lengths in *Snodgrassella* isolates from *Apis mellifera* (here referred to as Apis group2) than those in isolates from *Bombus* spp. (Fig. 2). The latter represented five lineages (Fig. 2 and Table 1). The first, Bombus group1, was composed of four isolates from *Bombus pensylvanicus* (all collected in the United States) and was fully supported by the bootstrap and leave-one-out analyses (Fig. S2 and S3). The second, Bombus group2, consisted of a single isolate from *Bombus nevadensis* (United States). The three remaining groups were fully supported in the bootstrap and leave-one-out analyses as well and were referred to as Bombus group3, which was composed of three isolates from *Bombus appositus* (United States), Bombus group4, which was composed of two isolates from *Bombus pascuorum* and *Bombus lapidarius* (Belgium), and Bombus group5, which comprised 17 isolates collected from 11 *Bombus*

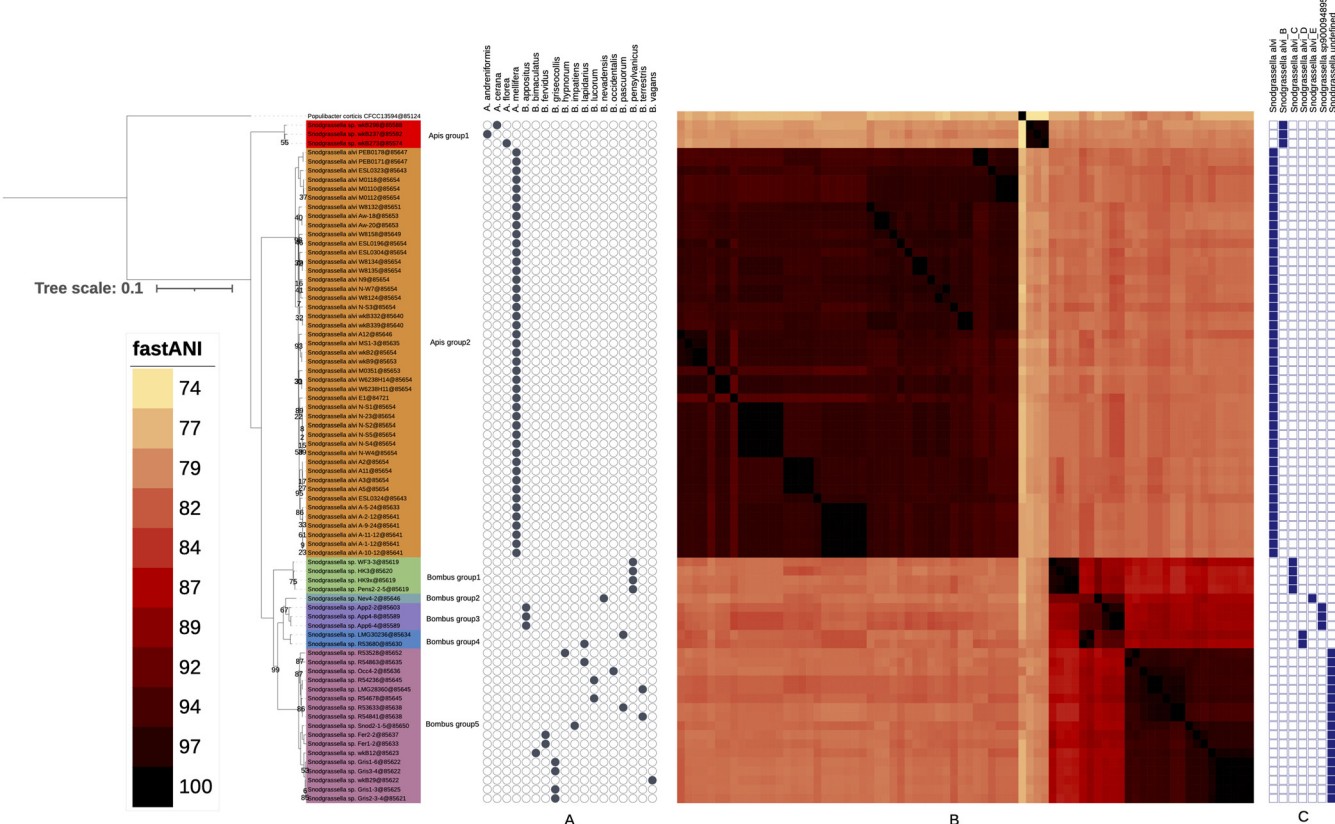

**FIG 2** *Snodgrassella* phylogeny and ANI comparison. A maximum likelihood tree was inferred from 254 core gene sequences under the PROTGAMMALGF model with RAxML (71) from a supermatrix of 76 organisms by 86,654 unambiguously aligned amino acid positions. Only bootstrap values below 100 are shown at the nodes. (A) Host. Host species were taken from biosample metadata in the NCBI portal, except for the 9 newly sequenced genomes, for which the metadata were taken from reference 15. (B) ANI heat map. ANI values were computed with fastANI (74). A triangular matrix was constructed according to pairwise distances. Colors associated with ANI values are given in the fastANI key. (C) GTDB hits. The blue squares on the right were determined using GTDBtk on the genomes of the associated strains (77).

species in Belgium and the United States (Fig. 2 and Table 1) and which included the isolates wkB12 and wkB29, reported by Kwong and Moran (21). This topology was confirmed with sparser taxon sampling, after removing 40 strains with dRep (43) and computing another large phylogenomic analysis (Fig. 3). The branching patterns of the seven groups were the same in all amino acid- and nucleotide-based trees inferred, except for Bombus group2. Bombus group2 appeared as a sister group of Bombus group3 and Bombus group4 in all analyses (with a bootstrap of <70), except in the nucleotide-based phylogenomic analysis, in which Bombus group2 branched with Bombus group1 (Fig. S3).

To quantify the taxonomic divergence between the observed phylogenomic groups, we performed ANI analyses. The latter are generally used for species delimitation in bacterial taxonomy (44), with ANI values of about 95 to 96% corresponding to the species delineation threshold (45, 46). The ANI analysis (Fig. 2, part ANI) revealed values within each of the phylogenomic groups (except Bombus group2, for which only a single genome was available) that were consistently above 95% ANI, while ANI values between genome sequences of different groups were consistently below 87%. Therefore, each of the seven phylogenomic groups corresponded to a distinct *Snodgrassella* species in modern bacterial taxonomy. These seven species also corresponded with the seven *Snodgrassella* species defined in the GTDB database (https://gtdb.ecogenomic.org/searches?s=al&q=Snodgrassella) (47) (Fig. 2, GTDB part). Additional *Snodgrassella* species likely exist as gut symbionts in stingless bees and *Bombus* species from as yet poorly examined continents, as suggested by their SSU rRNA phylogeny (10). The *S. alvi* strain wkB2$^T$ clustered within Apis group2, and by taxonomic convention, the name *S. alvi* should therefore be restricted to organisms from Apis group2. All other phylogenomic groups detected in the present and in earlier studies represent novel *Snodgrassella*

**TABLE 1** Details of the Snodgrassella strains and public assemblies[a]

| Genome accession no. | Strain no.[b] | Species | Group | Isolation source | Geographic origin | Completeness (%) | Contamination (%) | Length | No. of contigs (nt) | $N_{50}$ |
|---|---|---|---|---|---|---|---|---|---|---|
| GCF_002777775.1 | wkB237* | Snodgrassella sp. nov. | Apis group1 | A. andreniformis | Singapore | 99.57 | 0.85 | 2,321,012 | 21 | 327,688 |
| GCF_002777855.1 | wkB298* | Snodgrassella sp. nov. | Apis group1 | A. cerana | Singapore | 99.57 | 0.85 | 2,338,271 | 42 | 300,195 |
| GCF_002777655.1 | wkB273* | Snodgrassella sp. nov. | Apis group1 | A. florea | Singapore | 99.57 | 0.85 | 2,317,511 | 31 | 335,346 |
| GCF_013344955.1 | ESL0324 | Snodgrassella alvi | Apis group2 | A. mellifera | USA | 99.57 | 0.85 | 2,599,151 | 21 | 554,030 |
| GCF_013344995.1 | ESL0304 | Snodgrassella alvi | Apis group2 | A. mellifera | USA | 99.57 | 0.85 | 2,427,362 | 23 | 239,422 |
| GCF_013346865.1 | ESL0323 | Snodgrassella alvi | Apis group2 | A. mellifera | USA | 99.57 | 0.85 | 2,665,387 | 37 | 227,440 |
| GCF_016100435.1 | W8158 | Snodgrassella alvi | Apis group2 | A. mellifera | USA | 99.57 | 1.50 | 2,572,927 | 22 | 817,650 |
| GCF_016100485.1 | W6238H14 | Snodgrassella alvi | Apis group2 | A. mellifera | USA | 99.57 | 1.07 | 2,493,315 | 33 | 676,521 |
| GCF_016100525.1 | W6238H11 | Snodgrassella alvi | Apis group2 | A. mellifera | USA | 99.57 | 1.07 | 2,488,149 | 29 | 680,931 |
| GCF_016100575.1 | M0351 | Snodgrassella alvi | Apis group2 | A. mellifera | USA | 99.57 | 1.07 | 2,445,778 | 24 | 1,728,374 |
| GCF_016100865.1 | W8132 | Snodgrassella alvi | Apis group2 | A. mellifera | USA | 99.57 | 1.07 | 2,402,212 | 17 | 168,5768 |
| GCF_016100875.1 | W8124* | Snodgrassella alvi | Apis group2 | A. mellifera | USA | 99.57 | 0.85 | 2,374,073 | 13 | 1,777,221 |
| GCF_016101225.1 | W8135 | Snodgrassella alvi | Apis group2 | A. mellifera | USA | 99.57 | 0.85 | 2,403,944 | 18 | 1,368,577 |
| GCF_016101245.1 | W8134 | Snodgrassella alvi | Apis group2 | A. mellifera | USA | 99.57 | 0.85 | 2,404,985 | 18 | 1,368,577 |
| GCF_016101465.1 | M0110 | Snodgrassella alvi | Apis group2 | A. mellifera | USA | 99.57 | 0.64 | 2,542,136 | 49 | 358,478 |
| GCF_016101495.1 | M0112 | Snodgrassella alvi | Apis group2 | A. mellifera | USA | 99.57 | 0.64 | 2,542,583 | 47 | 358,478 |
| GCF_016101535.1 | M0118 | Snodgrassella alvi | Apis group2 | A. mellifera | USA | 99.57 | 0.64 | 2,544,967 | 53 | 358,478 |
| GCF_000600005.1 | wkB2* | Snodgrassella alvi | Apis group2 | A. mellifera | USA | 99.57 | 0.85 | 2,527,978 | 1 | 2,527,978 |
| GCF_002088655.1 | N-23 | Snodgrassella alvi | Apis group2 | A. mellifera | Norway | 99.57 | 0.85 | 2,421,693 | 128 | 38,667 |
| GCF_002088575.1 | N9* | Snodgrassella alvi | Apis group2 | A. mellifera | Norway | 99.57 | 0.85 | 2,403,335 | 129 | 42,232 |
| GCF_002088755.1 | N-S1 | Snodgrassella alvi | Apis group2 | A. mellifera | Norway | 99.57 | 0.85 | 2,420,873 | 98 | 50,074 |
| GCF_002088595.1 | N-S2 | Snodgrassella alvi | Apis group2 | A. mellifera | Norway | 99.57 | 0.85 | 2,421,229 | 73 | 61,427 |
| GCF_002088635.1 | N-S3* | Snodgrassella alvi | Apis group2 | A. mellifera | Norway | 99.57 | 0.43 | 2,463,518 | 79 | 70,149 |
| GCF_002088665.1 | N-S4* | Snodgrassella alvi | Apis group2 | A. mellifera | Norway | 99.57 | 0.85 | 2,421,486 | 38 | 133,936 |
| GCF_002088765.1 | N-S5 | Snodgrassella alvi | Apis group2 | A. mellifera | Norway | 99.57 | 0.85 | 2,417,615 | 77 | 56,721 |
| GCF_002088525.1 | N-W4 | Snodgrassella alvi | Apis group2 | A. mellifera | Norway | 99.57 | 0.85 | 2,421,251 | 75 | 55,544 |
| GCF_002088555.1 | N-W7* | Snodgrassella alvi | Apis group2 | A. mellifera | Norway | 99.57 | 0.85 | 2,423,318 | 62 | 70,113 |
| GCF_002777865.1 | PEB0171* | Snodgrassella alvi | Apis group2 | A. mellifera | USA | 99.57 | 1.28 | 2,520,622 | 77 | 117,138 |
| GCF_002777875.1 | PEB0178 | Snodgrassella alvi | Apis group2 | A. mellifera | USA | 99.15 | 1.37 | 2,524,932 | 135 | 43,076 |
| GCF_002777735.1 | wkB9 | Snodgrassella alvi | Apis group2 | A. mellifera | USA | 99.57 | 0.88 | 2,548,535 | 15 | 663,898 |
| GCF_002088405.1 | A-10-12 | Snodgrassella alvi | Apis group2 | A. mellifera | USA | 99.57 | 0.43 | 2,501,655 | 63 | 65,004 |
| GCF_002089015.1 | A11 | Snodgrassella alvi | Apis group2 | A. mellifera | USA | 99.57 | 0.85 | 2,430,265 | 122 | 42,222 |
| GCF_002088695.1 | A-11-12 | Snodgrassella alvi | Apis group2 | A. mellifera | USA | 99.57 | 0.43 | 2,500,985 | 90 | 53,637 |
| GCF_002088735.1 | A-1-12* | Snodgrassella alvi | Apis group2 | A. mellifera | USA | 99.57 | 0.43 | 2,502,286 | 58 | 91,493 |
| GCF_002088675.1 | A12* | Snodgrassella alvi | Apis group2 | A. mellifera | USA | 99.57 | 0.85 | 2,399,919 | 214 | 19,825 |
| GCF_002088395.1 | A2* | Snodgrassella alvi | Apis group2 | A. mellifera | USA | 99.57 | 0.85 | 2,425,186 | 84 | 56,386 |
| GCF_002088585.1 | A-2-12 | Snodgrassella alvi | Apis group2 | A. mellifera | USA | 99.57 | 0.43 | 2,502,682 | 75 | 64,388 |
| GCF_002088455.1 | A3 | Snodgrassella alvi | Apis group2 | A. mellifera | USA | 99.57 | 0.85 | 2,428,731 | 109 | 44,108 |
| GCF_002088475.1 | A5 | Snodgrassella alvi | Apis group2 | A. mellifera | USA | 99.57 | 0.85 | 2,430,376 | 120 | 40,275 |
| GCF_002088465.1 | A-5-24 | Snodgrassella alvi | Apis group2 | A. mellifera | USA | 99.57 | 0.43 | 2,490,743 | 172 | 28,499 |
| GCF_002088515.1 | A-9-24 | Snodgrassella alvi | Apis group2 | A. mellifera | USA | 99.57 | 0.43 | 2,501,107 | 62 | 83,544 |
| GCF_002088415.1 | Aw-18 | Snodgrassella alvi | Apis group2 | A. mellifera | USA | 99.57 | 0.85 | 2,497,111 | 88 | 55,786 |
| GCF_002019415.1 | Aw-20* | Snodgrassella alvi | Apis group2 | A. mellifera | USA | 99.57 | 0.85 | 2,498,497 | 65 | 75,805 |
| GCF_003202885.1 | ESL0196* | Snodgrassella alvi | Apis group2 | A. mellifera | Switzerland | 99.57 | 0.43 | 2,446,304 | 15 | 1,281,809 |
| GCF_002777925.1 | MS1-3 | Snodgrassella alvi | Apis group2 | A. mellifera | USA | 99.57 | 0.85 | 2,500,663 | 93 | 50,239 |

**TABLE 1** (Continued)

| Genome accession no. | Strain no.[b] | Species | Isolation source | Group | Geographic origin | Completeness (%) | Contamination (%) | Length | No. of contigs (nt) | $N_{50}$ |
|---|---|---|---|---|---|---|---|---|---|---|
| GCF_002777695.1 | wkB332* | Snodgrassella alvi | A. mellifera | Apis group2 | Malaysia | 99.57 | 1.28 | 2,487,419 | 30 | 502,922 |
| GCF_002777815.1 | wkB339 | Snodgrassella alvi | A. mellifera | Apis group2 | Malaysia | 99.57 | 0.85 | 2,499,917 | 27 | 431,105 |
| GCF_002406645.1 | E1* | Snodgrassella alvi | A. mellifera | Apis group2 | USA | 98.24 | 0.85 | 2,388,322 | 156 | 47,816 |
| GCF_002777705.1 | HK3* | Snodgrassella sp. nov. | B. pensylvanicus | Bombus group1 | USA | 99.57 | 0.90 | 2,609,176 | 74 | 92,612 |
| GCF_002777795.1 | HK9x* | Snodgrassella sp. nov. | B. pensylvanicus | Bombus group1 | USA | 99.57 | 0.43 | 2,568,638 | 110 | 120,216 |
| GCF_002777325.1 | Pens2-2-5 | Snodgrassella sp. nov. | B. pensylvanicus | Bombus group1 | USA | 99.57 | 0.85 | 2,486,143 | 132 | 102,952 |
| GCF_002777745.1 | WF3-3* | Snodgrassella sp. nov. | B. pensylvanicus | Bombus group1 | USA | 99.57 | 1.28 | 2,603,503 | 67 | 133,888 |
| GCF_002777825.1 | Nev4-2* | Snodgrassella sp. nov. | B. nevadensis | Bombus group2 | USA | 91.45 | 0.00 | 2,582,966 | 90 | 95,658 |
| GCF_002777315.1 | App2-2* | Snodgrassella sp. nov. | B. appositus | Bombus group3 | USA | 99.15 | 0.43 | 2,483,892 | 119 | 47,667 |
| GCF_002777425.1 | App4-8 | Snodgrassella sp. nov. | B. appositus | Bombus group3 | USA | 99.57 | 0.85 | 2,577,724 | 115 | 71,597 |
| GCF_002777465.1 | App6-4* | Snodgrassella sp. nov. | B. appositus | Bombus group3 | USA | 99.57 | 0.85 | 2,623,320 | 108 | 97,352 |
| GCA_914768095 | R-53680* | Snodgrassella sp. nov. | B. lapidarius | Bombus group4 | Belgium | 99.57 | 0.43 | 2,526,951 | 46 | 155,676 |
| GCA_914768025 | LMG 30236* | Snodgrassella sp. nov. | B. pascuorum | Bombus group4 | Belgium | 99.57 | 0.00 | 2,507,157 | 45 | 132,469 |
| GCF_000695565.1 | wkB12* | Snodgrassella sp. nov. | B. bimaculatus | Bombus group5 | USA | 99.57 | 0.43 | 2,438,497 | 34 | 337,275 |
| GCF_002777415.1 | Fer1-2* | Snodgrassella sp. nov. | B. fervidus | Bombus group5 | USA | 99.57 | 0.85 | 2,442,708 | 66 | 153,029 |
| GCF_002777485.1 | Fer2-2* | Snodgrassella sp. nov. | B. fervidus | Bombus group5 | USA | 99.57 | 0.85 | 2,358,796 | 53 | 132,767 |
| GCF_002777335.1 | Gris2-3-4 | Snodgrassella sp. nov. | B. griseocollis | Bombus group5 | USA | 99.57 | 0.43 | 2,417,520 | 62 | 183,509 |
| GCF_002777525.1 | Gris1-3 | Snodgrassella sp. nov. | B. griseocollis | Bombus group5 | USA | 99.57 | 0.48 | 2,441,972 | 58 | 128,979 |
| GCF_002777615.1 | Gris1-6 | Snodgrassella sp. nov. | B. griseocollis | Bombus group5 | USA | 99.57 | 0.43 | 2,459,690 | 50 | 149,104 |
| GCF_002777595.1 | Gris3-4 | Snodgrassella sp. nov. | B. griseocollis | Bombus group5 | USA | 99.57 | 0.43 | 2,453,208 | 48 | 122,377 |
| GCA_914768055 | R-53528* | Snodgrassella sp. nov. | B. hypnorum | Bombus group5 | Belgium | 99.57 | 0.43 | 2,287,013 | 43 | 179,794 |
| GCF_002777345.1 | Snod2-1-5* | Snodgrassella sp. nov. | B. impatiens | Bombus group5 | USA | 99.57 | 0.43 | 2,355,707 | 62 | 146,579 |
| GCA_914768015 | R-54863* | Snodgrassella sp. nov. | B. lapidarius | Bombus group5 | Belgium | 99.57 | 0.43 | 2,434,137 | 60 | 124,074 |
| GCA_914768085 | R-54236 | Snodgrassella sp. nov. | B. lucorum | Bombus group5 | Belgium | 99.57 | 0.43 | 2,314,670 | 26 | 184,956 |
| GCA_914768035 | R-54678* | Snodgrassella sp. nov. | B. lucorum | Bombus group5 | Belgium | 99.57 | 0.43 | 2,265,685 | 32 | 179,671 |
| GCF_002777635.1 | Occ4-2* | Snodgrassella sp. nov. | B. occidentalis | Bombus group5 | USA | 99.57 | 0.43 | 2,487,519 | 54 | 151,012 |
| GCA_914768065 | R-53633* | Snodgrassella sp. nov. | B. pascuorum | Bombus group5 | Belgium | 99.57 | 0.43 | 2,402,464 | 42 | 196,721 |
| GCA_914768075 | R-54841 | Snodgrassella sp. nov. | B. terrestris | Bombus group5 | Belgium | 99.57 | 0.43 | 2,400,984 | 40 | 170,243 |
| GCA_914768045 | LMG 28360* | Snodgrassella sp. nov. | B. terrestris | Bombus group5 | Belgium | 99.57 | 0.43 | 2,310,392 | 28 | 184,956 |
| GCF_000695545.1 | wkB29 | Snodgrassella sp. nov. | B. vagans | Bombus group5 | USA | 99.57 | 0.43 | 2,398,206 | 88 | 184,072 |
| GCF_001590725.1 | CFCC 13594* | Populibacter corticis | P. canker | Outgroup | USA | 98.72 | 0.00 | 2,372,914 | 131 | 95,649 |

[a]Completeness and contamination values/metrics were estimated with CheckM (61). Assembly statistics were computed with QUAST (62).
[b]Asterisks indicate the strains retained by dRep after dereplication.

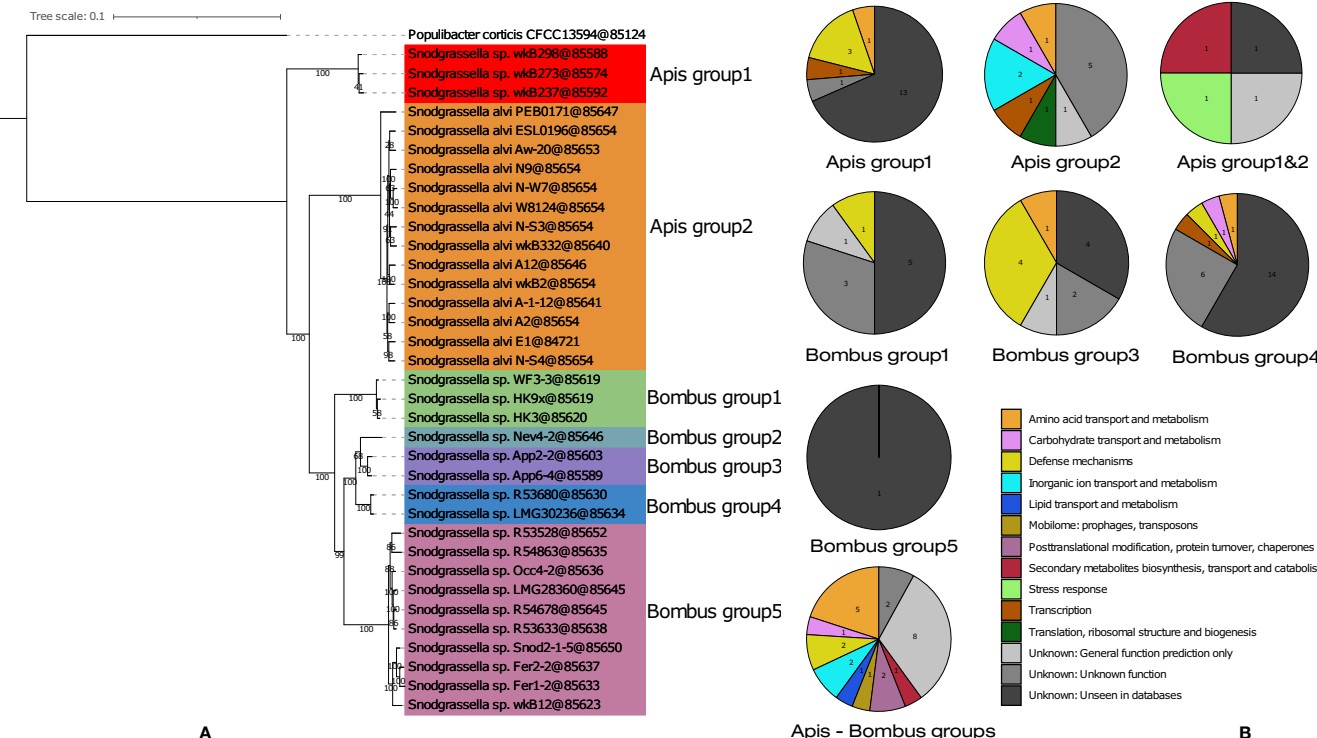

**FIG 3** *Snodgrassella* phylogeny, after dereplication of highly similar genomes, and metabolic analysis per species group. (A) Maximum likelihood tree inferred on 254 core genes under the PROTGAMMALGF model with RAxML (71) from a supermatrix of 36 organisms by 86,654 unambiguously aligned amino acid positions. Bootstrap support values are shown at the nodes. (B) Functional analyses were performed using COG (68) and Mantis (53). Numbers indicated in the pie charts correspond to absolute numbers of OGs identified in the respective *Snodgrassella* subgroups. Specific genes were computed for entire groups before dereplication.

species. Below we propose to formally name Bombus group4 as *Snodgrassella gandavensis*, with LMG 30236 (=CECT 30450) as the type strain, and Bombus group5 as *Snodgrassella communis*, with LMG 28360 (=CECT 30451) as the type strain. The formal naming of the remaining phylogenomic groups (i.e., Apis group1, Bombus group1, Bombus group2, and Bombus group3) can be done upon characterization and public deposit of reference cultures to conform to standard practices in bacterial taxonomy (44).

It has been suggested that the transmission mode of *Snodgrassella* between workers and larvae is mainly vertical within a colony (27, 32, 48). Although for most of the groups (i.e., *Snodgrassella* species) detected in the present study, only a limited number of isolates or genomes were available, our data showed that indeed several *Snodgrassella* species occur in more than one *Bombus* species, and therefore this rejects strict cospeciation. This result is in agreement with data reported by Powel et al. (42), who demonstrated that Bombus group5 *Snodgrassella* (i.e., *S. communis*) was detected in several *Bombus* species, and with the experimental demonstration of host jumps in *Bombus* species (2). The close phylogenetic relationship between *Snodgrassella* species associated with bumblebees and *Snodgrassella* species associated with the Western honeybee additionally supports the hypothesis of horizontal transfer of symbionts between bee clades. The origin of extant bumblebees is estimated to the Miocene, as based on fossil records and molecular phylogeny (49, 50). The origin of this clade is associated with a global cooling of the Palearctic region (51). Ancestors of bumblebees were likely sharing host plants with the Western honeybee ancestor clade, which is also associated with a temperate climate, and not with the *Apis cerana* honeybee clade, which is associated with a subtropical to tropical warmer climate. Overall, the similarity of microbiota among corbiculate bees seems to be related to both phylogeny and sharing of common habitats/climates.

**Specific gene analysis.** *Snodgrassella* has been coevolving with other bacteria in the gut of honeybees and bumblebees for 80 million years (9, 10, 21). Interestingly the age of the common ancestor of the clade of the hosts (i.e., corbiculate bees, which include both

bumblebees and honeybees) is also estimated round ~85 million years (52). Different species and genotypes have evolved and may have developed different functions in their hosts. We investigated the presence of group-specific genes (i.e., genes that are exclusively present in one group [or species] and that are shared by all members of this group) and determined their functionalities. The rationale was that a conserved gene probably confers a benefit to members of that group. We detected 107 specific genes: 19 were specific for Apis group1, 12 for Apis group2 (i.e., *S. alvi*), 10 for Bombus group1, 12 for Bombus group3, 24 for Bombus group4 (*S. gandavensis*), and 1 for Bombus group5 (*S. communis*) (Fig. 3). No specific genes were found for Bombus group2. In a next step, genes shared by multiple groups or species were examined. Only two group combinations presented specific genes: Apis group1 and Apis group2 shared 4 genes that were absent in all others, and all *Snodgrassella* spp. together shared 25 genes (Fig. 3).

Of these 107 genes, 37 did not correspond to proteins known in COG or Mantis (53). The latter is the most recent and comprehensive tool for protein annotation and contains Pfam, eggnog, NPFM, TIGRfams, and Kofam family data for the three domains of life (53). These 37 proteins were relatively short (median, 53 amino acids [aa]; interquartile range [IQR], 46 aa) compared to the rest of the 107 genes (median, 197 aa; IQR, 202.25), with only six proteins above 100 aa (Table 2). Small proteins in bacteria can have roles in transport and signal transduction, can act as chaperones, can be involved in stress responses or virulence (54, 55), and can also be used as bacteriocins (56). Unknown genes are a recurrent issue in metagenomic studies, where they may represent up to 40% of the genes (57). Besides these genes without a match, 24 genes encoded proteins with unknown function but with hits in COG or Mantis, while six genes had a general predicted function only and could not be affiliated with a metabolic pathway. Together, 67 of 107 specific genes (62.6%) and their putative benefits remained unknown (Table 2 and Fig. 3).

Eleven of the 40 identified specific genes represented defense mechanisms, which are important for gut colonization in bees (14) (Table 2 and Fig. 3). The two most conserved defense mechanisms (4 genes each) corresponded to drug exporter and Rhs proteins (Table 2). These two defense mechanisms generate similar benefits and can provide competitive advantages. Steele and Moran (34) recently demonstrated that Rhs proteins represent different toxins, which are injected in neighboring cells through a type VI secretion system (T6SS) (2, 32–34). The four Rhs proteins found in the present study were specific genes of Bombus group3, a gut symbiont thus far detected only in *B. appositus*. The presence of specific toxins in these three strains might explain the specificity for their host by T6SS mediated-competition. The last three defense mechanisms included proteins for glycosylation of phage DNA (Bombus group1), sensing of invasive viruses, and antibiotic production (Apis group1) (Table 2). Apis group1 possessed a specific protein closely related to the 2-oxo-3-(phosphooxy)propyl 3-oxoalkanoate synthase, which triggers the production of the antibiotic virginiamycin in *Streptomyces virginiae* (58).

Among the remaining identified specific genes, 9 genes belonged to the metabolic category amino acid transport and metabolism, 4 genes belonged to inorganic ion transport and metabolism, and 4 genes belonged to carbohydrate transport and metabolism (Fig. 3). Because host gut environments are scarce in iron and amino acids, *de novo* biosynthetic pathways and iron importers have been reported to be essential for gut host colonization (59). Specific genes linked to carbohydrate metabolism might provide a benefit linked to a carbohydrate-rich diet (1), even if such genes are more commonly found in *Gilliamella* than *Snodgrassella* genomes (30). The 12 remaining identified specific genes represented different metabolic pathways with unclear function in, or benefit to, the host (Table 2).

**Horizontal gene transfer.** Horizontal gene transfer is a process that plays a major role in bacterial evolution. Because of the spatial proximity of *Snodgrassella* and other gut bacteria during 80 million years of coevolution, HGT events may have impacted the specific gene complement. We investigated the potential presence of such events by using MetaCHIP (60). We included all bee gut symbiont genomes available in RefSeq (561 genomes: i.e., 75 of *Snodgrassella* plus 486 of representatives of other bee gut symbionts) to maximize the odds of HGT detection and detected only 57 events of HGT in which *Snodgrassella* spp. were a donor or recipient (Fig. S5 and Table S3). This is less than the 87 events reported by Kwong et al. (2).

**TABLE 2** Functional analysis of the specific genes[a]

| Group | Gene ID[b] | Length (aa) | Gene product | Mapping tool | COG pathway | Function |
|---|---|---|---|---|---|---|
| Apis group1 | GC_00003784 | 100 | | | Unknown | |
| | GC_00004036 | 330 | AfsA | Mantis | Defense mechanisms | 2-Oxo-3-(phosphooxy)propyl 3-oxoalkanoate synthase |
| | GC_00004009 | 235 | GlnQ | COG | Amino acid transport and metabolism | ABC-type polar amino acid transport system, ATPase component (GlnQ) (PDB no. 4YMS) |
| | GC_00003806 | 86 | | | Unknown | |
| | GC_00003947 | 43 | | | Unknown | |
| | GC_00003950 | 83 | | Mantis | Defense mechanisms | Permeases of drug metabolite transporter (DMT) superfamily |
| | GC_00003860 | 32 | | | Unknown | |
| | GC_00003920 | 32 | | | Unknown | |
| | GC_00003767 | 110 | | | Unknown | |
| | GC_00004005 | 86 | | | Unknown | |
| | GC_00004028 | 78 | | | Unknown | |
| | GC_00003774 | 33 | | Mantis | Function unknown | |
| | GC_00003768 | 98 | SoxR | COG | Transcription | DNA-binding transcriptional regulator, MerR family (SoxR) (PDB no. 2VZ4) |
| | GC_00003846 | 53 | | | Unknown | |
| | GC_00004043 | 69 | | | Unknown | |
| | GC_00004011 | 39 | vWA-MoxR | Mantis | Defense mechanisms | vWA-MoxR-associated protein middle region (VMAP-M) 1—sensing of invasive entities (EMBL) |
| | GC_00003887 | 53 | | | Unknown | |
| | GC_00004656 | 72 | | | Unknown | |
| | GC_00003916 | 47 | | | Unknown | |
| Apis group2 | GC_00001836* | 230 | | COG | Function unknown | Uncharacterized protein, contains DUF2461 domain |
| | GC_00001864 | 244 | | Mantis | Function unknown | Uncharacterized protein |
| | GC_00001813 | 75 | YozG | COG | Transcription | DNA-binding transcriptional regulator, XRE family (YozG) (PDB no. 3TYR) |
| | GC_00001812* | 213 | PspE | COG | Inorganic ion transport and metabolism | Rhodanese-related sulfurtransferase (PspE) (PDB no. 1TQ1) |
| | GC_00001818* | 598 | LepA | COG | Translation, ribosomal structure and biogenesis | Translation elongation factor EF-4, membrane-bound GTPase (LepA) (PDB no. 3DEG) |
| | GC_00001830 | 114 | | Mantis | Function unknown | |
| | GC_00001846 | 394 | AraJ | COG | Carbohydrate transport and metabolism | Predicted arabinose efflux permease AraJ, MFS family (AraJ) (PDB no. 4LDS) |
| | GC_00001829 | 101 | MdaB | COG | General function prediction only | Putative NADPH-quinone reductase (modulator of drug activity B) (MdaB) (PDB no. 1D4A) |
| | GC_00001849 | 289 | | Mantis | Function unknown | |
| | GC_00001848 | 355 | Lys9 | COG | Amino acid transport and metabolism | Saccharopine dehydrogenase, NADP-dependent (Lys9) (PDB no. 1E5L) |
| | GC_00001833 | 272 | FTR1 | COG | Inorganic ion transport and metabolism | High-affinity $Fe^{2+}/Pb^{2+}$ permease (FTR1) |
| | GC_00001824* | 180 | | Mantis | Function unknown | |
| Apis group1&2 | GC_00001778 | 262 | YaaA | Mantis | Stress response | Peroxide stress protein YaaA |
| | GC_00001768 | 183 | | | Unknown | |
| | GC_00001766 | 167 | PadC | COG | Secondary metabolites biosynthesis, transport and catabolism | Phenolic acid decarboxylase (PadC) (PDB no. 2GC9) |
| | GC_00001776 | 328 | | Mantis | General function prediction only | Short C-terminal domain |
| Bombus group1 | GC_00003663 | 110 | MSP7_C | Mantis | General function prediction only | MSP7-like protein C-terminal domain |
| | GC_00003592 | 76 | | Mantis | Function unknown | |
| | GC_00003535 | 31 | | | Unknown | |
| | GC_00003491 | 70 | GmrSD | COG | Defense mechanisms | DNAse/DNA nickase specific for phosphorothioated or glycosylated phage DNA, GmrSD/DndB/SspE family |
| | GC_00003648 | 62 | | Mantis | Function unknown | |

**TABLE 2** (Continued)

| Group | Gene ID[b] | Length (aa) | Gene product | Mapping tool | COG pathway | Function |
|---|---|---|---|---|---|---|
| | GC_00003489 | 29 | | | Unknown | |
| | GC_00003623 | 29 | | | Unknown | |
| | GC_00003493 | 35 | | | Unknown | |
| | GC_00003540 | 69 | | | Unknown | |
| | GC_00003502 | 41 | | Mantis | Function unknown | |
| Bombus group3 | GC_00004090 | 277 | | Mantis | Defense mechanisms | Rhs family protein |
| | GC_00003889 | 86 | DHAD | Mantis | Amino acid transport and metabolism | Dihydroxy-acid dehydratase |
| | GC_00003888 | 115 | | | Unknown | |
| | GC_00003882 | 62 | | Mantis | Function unknown | |
| | GC_00003783 | 69 | | | Unknown | |
| | GC_00004096 | 680 | | Mantis | Defense mechanisms | Rhs family protein |
| | GC_00003945 | 52 | | | Unknown | |
| | GC_00003718 | 266 | | Mantis | Defense mechanisms | Rhs family protein |
| | GC_00003802 | 94 | | | Unknown | |
| | GC_00004066 | 361 | | Mantis | Defense mechanisms | Rhs family protein |
| | GC_00003778 | 83 | | Mantis | Function unknown | |
| | GC_00003796 | 71 | | Mantis | General function prediction only | Deoxyhypusine monooxygenase activity |
| Bombus group4 | GC_00004382 | 166 | | Mantis | Function unknown | |
| | GC_00004653 | 110 | | Mantis | Function unknown | |
| | GC_00004442 | 38 | | | Unknown | |
| | GC_00004629 | 47 | | | Unknown | |
| | GC_00004489 | 36 | | | Function unknown | |
| | GC_00004467 | 254 | MMPL | COG | Defense mechanisms | Predicted exporter protein, RND superfamily (MMPL) (PDB no. 5KHN) drug exporter |
| | GC_00004294* | 395 | AraJ | COG | Carbohydrate transport and metabolism | Predicted arabinose efflux permease AraJ, MFS family (AraJ) (PDB no. 4LDS) |
| | GC_00004680 | 160 | | | Unknown | |
| | GC_00004725 | 34 | | | Unknown | |
| | GC_00004304 | 69 | Fic_N | COG | Transcription | Fic family protein (PDB no. 3CUC) |
| | GC_00004186 | 46 | | | Unknown | |
| | GC_00004560* | 62 | HIS2 | COG | Amino acid transport and metabolism | Histidinol phosphatase or related hydrolase of the PHP family (HIS2) (PDB no. 1M65) |
| | GC_00004631 | 33 | | | Unknown | |
| | GC_00004357 | 80 | | | Unknown | |
| | GC_00004742 | 64 | | | Unknown | |
| | GC_00004735 | 30 | | | Unknown | |
| | GC_00004255 | 104 | | Mantis | Function unknown | |
| | GC_00004450 | 61 | | | Unknown | |
| | GC_00004308 | 72 | | Mantis | Function unknown | |
| | GC_00004249 | 38 | | Mantis | Function unknown | |
| | GC_00004340 | 39 | | | Unknown | |
| | GC_00004716 | 34 | | | Unknown | |
| | GC_00004446 | 33 | | | Unknown | |
| | GC_00004728 | 66 | | Mantis | Function unknown | |
| Bombus group5 | GC_00002418 | 120 | | | Unknown | |
| Apis-Bombus groups | GC_00001509 | 376 | | Mantis | Function unknown | |
| | GC_00001371 | 68 | | Mantis | Function unknown | |
| | GC_00001428* | 270 | NlpA | COG | Inorganic ion transport and metabolism | ABC-type metal ion transport system, periplasmic component/surface antigen (NlpA) (PDB no. 1P99) |
| | GC_00001520 | 548 | DAK1 | COG | Carbohydrate transport and metabolism | Dihydroxyacetone kinase (DAK1) (PDB no. 1UN8) |
| | GC_00001461 | 309 | DnaJ | COG | Posttranslational modification, protein turnover, chaperones | DnaJ-class molecular chaperone with C-terminal Zn finger domain (DnaJ) (PDB no. 1BQ0) |

**TABLE 2** (Continued)

| Group | Gene ID[b] | Length (aa) | Gene product | Mapping tool | COG pathway | Function |
|---|---|---|---|---|---|---|
| | GC_00001484 | 273 | YvaK | COG | Secondary metabolites biosynthesis, transport and catabolism | Esterase/lipase (YvaK) (PDB no. 4DIU) |
| | GC_00001382 | 118 | | Mantis | Function unknown | |
| | GC_00001454 | 299 | YjjU | COG | Lipid transport and metabolism | Predicted phospholipase, patatin/cPLA2 family (YjjU) |
| | GC_00001527 | 359 | | Mantis | Function unknown | |
| | GC_00001467 | 142 | | Mantis | Function unknown | |
| | GC_00001482 | 268 | NosY | COG | Posttranslational modification, protein turnover, chaperones | ABC-type transport system involved in multicopper enzyme maturation, permease component (NosY) |
| | GC_00001439 | 381 | | Mantis | Function unknown | |
| | GC_00001398 | 208 | | Mantis | Function unknown | |
| | GC_00001393 | 135 | EmrE | COG | Defense mechanisms | Multidrug transporter EmrE and related cation transporters (EmrE) (PDB no. 2I68) |
| | GC_00001469 | 144 | ElaA | COG | General function prediction only | Predicted *N*-acyltransferase, GNAT family (ElaA) (PDB no. 1XEB |
| | GC_00001463 | 296 | SpeB | COG | Amino acid transport and metabolism | Arginase/agmatinase family enzyme (SpeB) (PDB no. 1CEV) |
| | GC_00001414 | 118 | EmrE | COG | Defense mechanisms | Multidrug transporter EmrE and related cation transporters (EmrE) (PDB no. 2I68) |
| | GC_00001408* | 277 | NlpA | COG | Inorganic ion transport and metabolism | ABC-type metal ion transport system, periplasmic component/surface antigen (NlpA) (PDB no. 1P99) |
| | GC_00001441 | 142 | | Mantis | Function unknown | |
| | GC_00001474 | 186 | | Mantis | General function prediction only | Peptidase activity |
| | GC_00001539 | 248 | MtnX | COG | Amino acid transport and metabolism | 2-Hydroxy-3-keto-5-methylthiopentenyl-1-phosphate phosphatase (methionine salvage) (MtnX) (PDB no. 2FEA) |
| | GC_00001507 | 251 | | COG | Mobilome: prophages, transposons | Phage repressor protein C, contains Cro/C1-type HTH and peptidase s24 domains |
| | GC_00001374 | 530 | AspB | COG | Amino acid transport and metabolism | Aspartate/methionine/tyrosine aminotransferase (AspB) (PDB no. 2O0R) |
| | GC_00001419 | 471 | ArgD | COG | Amino acid transport and metabolism | Acetylornithine aminotransferase/4-aminobutyrate aminotransferase (ArgD) (PDB no. 1SF2) |
| | GC_00001464 | 225 | YaeF/YiiX | Mantis | Amino acid transport and metabolism | Permuted papain-like amidase enzyme, YaeF/YiiX, C92 family |

[a]Functional analyses were performed with anvi'o (67), using COGs (68) and Mantis (53). Pathways correspond to COG pathways as indicated by anvi'o. Unknown genes correspond to genes without hits (unseen in databases).
[b]Asterisks indicate putative HGT events.

MetaCHIP is designed to infer HGT from genomes assembled from microbial communities (60) and detects potential HGT events using a BLAST best-hit approach and then validates these hits by performing a duplication-transfer-loss (DTL) reconciliation between a species tree and an individual gene tree. The DTL filter used by MetaCHIP is more stringent than the BLAST E value threshold of 1e−50 used by Kwong et al. (2), which may explain this difference.

None of the 107 specific genes described above was present among the 57 HGT-affected genes detected by MetaCHIP using BLASTP and a filter on 98% identity (Table S3). We investigated the absence of HGT in the group of 107 specific genes further by enriching the comparison with orthologous sequences taken from the 561 gut symbiont genomes used above and with orthologous sequences from the 247 genomes unrelated to the bee gut ecosystem, thereby also covering bacterial and archaeal diversity (Fig. 4). This allowed us to analyze the taxonomic diversity of the specific genes and revealed that 68 genes were unique to *Snodgrassella*, while 39 genes were found in other bacteria too (27 genes in multiple other bacteria and 12 genes in only one other bacterium) (Fig. 4). The evolutionary history of the 68 genes unique to *Snodgrassella* is explained more easily by duplication in *Snodgrassella* than by acquisition by HGT or gene loss in all other bacteria. In contrast, the 27 specific genes found in other bee gut symbionts or in other bacteria unrelated to the bee gut

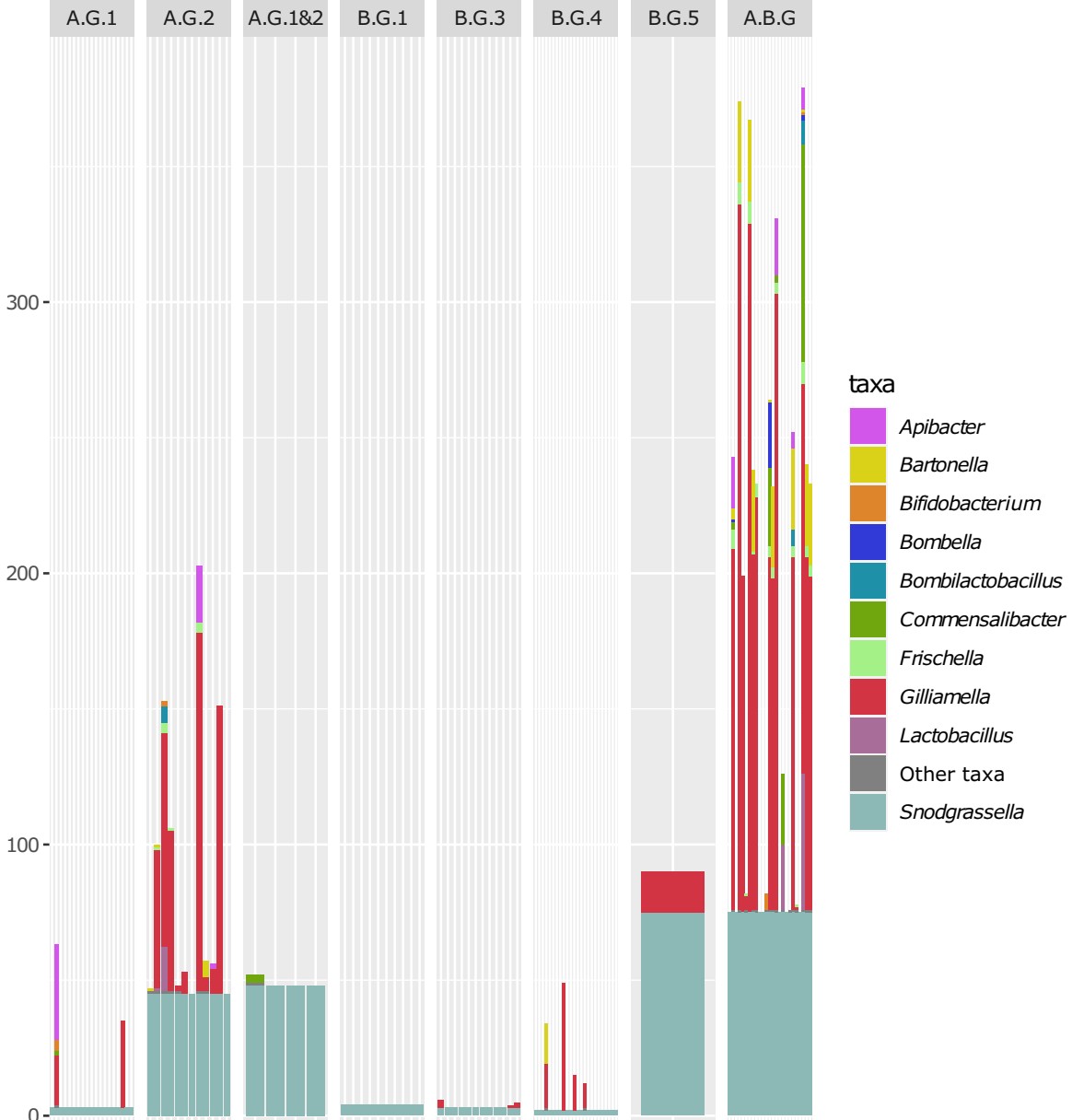

**FIG 4** Orthologous enrichment of group-specific genes. A total of 107 specific genes from bee gut bacteria (including 75 *Snodgrassella* genomes) were enriched with sequences from 486 genomes from bee gut bacteria and 247 genomes from bacteria not related to the bee gut ecosystem, using Forty-Two (75, 76). The *y* axis represents the absolute number of sequences, and the *x* axis represents individual genes. Genes were grouped by species group: A.G.1 for Apis group1 (19 genes), A.G.2 for Apis group2 (12 genes), A.G.1&2 for the two Apis groups together (4 genes), B.G.1 for Bombus group1 (10 genes), B.G.3 for Bombus group3 (12 genes), B.G.4 for Bombus group4 (24 genes), B.G.5 for Bombus group5 (1 gene), and A.B.G for Apis and Bombus groups together (25 genes).

microbiome suggest that gene loss in other *Neisseriaceae* was the most plausible evolutionary path of the genes. We computed individual gene trees for these 27 OGs (available at https://github.com/Lcornet/SNOD) and compared them manually to species trees (Fig. S6). For only 8 genes out of the 27 (indicated in Table 2), *Snodgrassella* sequences clustered with those of other bee gut symbionts, which may suggest HGT events. For 12 genes (out of 107), no convincing evidence of HGT was detected because these genes were only present in two species, making the comparison to the species tree impossible. Ten of the latter genes were detected in *Gilliamella* and *Snodgrassella* only, and two were detected in *Snodgrassella* and *Frischella* only. These genes included three out of four specific Rhs proteins of Bombus group3, which were shared with *Gilliamella* genomes.

**TABLE 3** Differential characteristics between *Snodgrassella alvi*, *Snodgrassella gandavensis*, and *Snodgrassella communis*

| | Result for[a]: | | | | | |
| | | *S. gandavensis* (Bombus group4) | | *S. communis* (Bombus group5) | | |
| Characteristic | *S. alvi* NCIMB 14803[T] | LMG 30236[T] | R-53680 | LMG 28360[T] | R-53528 | R-54236 |
|---|---|---|---|---|---|---|
| Growth on BHI agar | + | − | − | + | + | + |
| Growth on AC agar with 2% NaCl | ++ | w | w | ++ | + | + |
| Urease production | + | + | + | − | − | − |
| Growth on DNase agar | + | − | − | − | − | − |
| Hydrolysis of Tween 20 | − | + | + | − | − | − |

[a]+, present; −, absent; w, weak reaction.

**Conclusion.** In the present study, we used highly conserved *Neisseriaceae* core genes to perform multiple phylogenomic analyses. We demonstrated monophyly of *Snodgrassella* strains isolated from bumblebees and paraphyly of strains isolated from honeybees. We also demonstrated that *Snodgrassella* strains from Asian honeybees are an early diverging group. Combined with ANI analyses, our data further indicated that this genus comprises at least seven species. Below we describe and formally name two new *Snodgrassella* species from bumblebees: i.e., *S. gandavensis* sp. nov. for Bombus group4 (with LMG 30236 as the type strain) and *S. communis* sp. nov. for Bombus group5 (with LMG 28360 as the type strain). We detected 107 specific genes among these seven species. For the large majority of these 107 genes, there was no evidence for HGT. Functional analyses revealed the importance of small proteins, defense mechanisms, amino acid transport and metabolism, inorganic ion transport and metabolism, and carbohydrate transport and metabolism among these specific genes.

**Description of *Snodgrassella gandavensis* sp. nov.** *Snodgrassella gandavensis* (gan.da.-ven'sis. M.L. masc. adj. *gandavensis*, of *Gandavum*, the Latin name for Ghent, referring to the place where these bacteria were first isolated).

Cells are nonmotile, Gram stain-negative rods about 1.2 $\mu$m long and 0.8 $\mu$m wide that occur singly or in pairs. Optimal growth is on rich agar medium at 37°C in a $CO_2$-enriched atmosphere. *S. gandavensis* grows under anaerobic conditions when supplemented with 10 mM $KNO_3$, but not under aerobic conditions. The bacterium is positive for catalase activity and nitrate reduction and negative for oxidase activity.

The type strain is LMG 30236 (=CECT 30450), which was isolated in 2015 from the gut of *Bombus pascuorum* sampled in 2015 in Wetteren, Belgium. The whole-genome sequence of LMG 30236[T] has a size of 2.51 Mbp. The DNA G+C content is 43.76 mol%. The whole-genome sequence is publicly available under accession no. GCA_914768025.1. The 16S rRNA gene sequence is publicly available under accession no. OU943324.

Additional information is provided in Table 3 and the supplemental material.

**Description of *Snodgrassella communis* sp. nov.** *Snodgrassella communis* (com.-mu'nis. L. fem. adj. *communis*, common, because of its wide host range).

Cells are nonmotile, Gram stain-negative rods about 1.2 $\mu$m long and 0.8 $\mu$m wide that occur singly or in pairs. Optimal growth is on rich agar medium at 37°C in a $CO_2$-enriched atmosphere. *S. communis* grows under anaerobic conditions when supplemented with 10 mM $KNO_3$. Growth under aerobic conditions is strain dependent. The organism is positive for catalase activity and nitrate reduction and negative for oxidase activity.

The type strain is LMG 28360 (=CECT 30451), which was isolated in 2013 from the gut of *Bombus terrestris* sampled in 2013 in Ghent, Belgium. The whole-genome sequence of LMG 28360[T] has a size of 2.31 Mbp. The DNA G+C content is 43.26 mol%. The whole-genome sequence is publicly available under accession no. GCA_914068745.1. The 16S rRNA gene sequence is publicly available under accession no. OU943323.

Additional information is provided in Table 3 and the supplemental material.

## MATERIALS AND METHODS

All custom scripts specifically developed for this study are available at https://github.com/Lcornet/SNOD.

***Snodgrassella* whole-genome sequences.** Whole-genome sequences of 75 *Snodgrassella* isolates were analyzed. To this end, all *Snodgrassella* genomes from the Reference Sequence database of NCBI (RefSeq) (61, 62) were downloaded on 25 June 2021. CheckM v1.1.3 (63) with the *lineage_wf* option was

used to identify genomes with contamination levels below 5% and completeness above 95%, which yielded 66 RefSeq genomes. In addition, we determined whole-genome sequences of nine bumblebee isolates from an earlier study (15). Together, the genomes of 48 isolates originated from honeybee gut samples (from 4 species) and 27 from bumblebee gut samples (from 14 species). Assembly quality metrics for all 75 genomes were obtained with QUAST v5.0.2 (64). The isolates, along with their geographic origin, accession numbers, and CheckM and QUAST parameters, are listed in Table 1.

**DNA extraction.** Genomic DNA was isolated using a Maxwell 16 tissue DNA purification kit (catalog no. AS1030) and a Maxwell 16 instrument (catalog no. AS2000). The integrity and purity of the DNA were evaluated on a 1.0% (wt/vol) agarose gel and by spectrophotometric measurements at 234, 260, and 280 nm. A Quantus fluorometer and a QuantiFluor ONE double-stranded DNA (dsDNA) system (Promega Corporation, Madison, WI, USA) were used to estimate the DNA concentration.

**Library construction and genome sequencing.** Library preparation and whole-genome sequencing were performed by the Oxford Genomics Centre (University of Oxford, United Kingdom). Library preparation was performed using an adapted protocol of the NEB prep kit. Paired-end sequence reads (PE150) were generated using an Illumina NovaSeq 6000 platform (Illumina, Inc., USA).

**Genome assembly.** Quality checking and trimming of the raw sequence reads and *de novo* genome assembly were performed using the Shovill v1.1.0 pipeline (https://github.com/tseemann/shovill), which uses SPAdes v3.14.0 (65) as its core and which subsamples reads to a sequencing depth of 150×. Contigs shorter than 500 bp were excluded from the final assembly. Quality checking of the assembly was performed using QUAST v5.0.2 (64) and CheckM v1.1.3 (63).

**Core gene analysis.** ToRQuEMaDA (66), 10 June 2021 version, was used to select genomes representing the diversity of other *Neisseriaceae* (i.e., bacteria not belonging to the genus *Snodgrassella*). We selected 35 *Neisseriaceae* genomes using the following options of tqmd_cluster.pl: *dist-threshold of 0.90, kmer-size of 12 and max-round of 10 and CheckM v1.1.3 activated* (see Table S1 in the supplemental material). The 110 genomes were then imported into the pangenomic workflow of anvi'o v7.1 (67). The genomes were first loaded individually into anvi'o using the anvi-gen-contigs-database script with default options, and NCBI COGs (68) were associated with each genome database using the anvi-run-ncbi-cogs script with default options. The consolidated database was constructed using the anvi-gen-genomes-storage script with default options and anvi-pan-genome script with the following options: *min-occurrence of 2 and mcl-inflation of 10*. The protein sequences of all genomes, along with their COG annotations, were extracted from the anvi'o database using the anvi-get-sequences-for-gene-clusters script with default options and the anvi-summarize script with default options, respectively. Functional and geometric indices of anvi'o were computed using the anvi-compute-gene-cluster-homogeneity script with default options. Orthologous groups (OGs) were reconstructed from anvi'o output files using the custom script anvio_pan-to-OGs.py with default options. Finally, core genes were selected using the custom script anvio_OGs-filtration.py with the following options: *pfilter set to yes, fraction set to 1, unwanted orgs limit set to 0, cfilter set to yes, maxcopy set to 1, hfilter set to 1, maxfunctional index set to 0.8 and maxgeometricindex set to 0.8*. These settings resulted in the selection of 254 core genes out of 11,185 OGs, present in 100% of the 110 *Neisseriaceae* genomes.

**Phylogenomic analyses. (i) *Neisseriaceae* phylogeny.** The 254 protein OGs were aligned using the anvi'o workflow. Conserved sites were selected using BMGE v1.12 (69) with moderately severe settings (*entropy cut-off* = 0.5, *gap cut-off* = 0.2). A supermatrix of 110 organisms by 85,654 unambiguously aligned amino acid positions (0.15% missing character states) was generated using SCaFoS v1.30k (70), with default settings. A *Neisseriaceae* phylogenomic analysis was inferred using RAxML v8.1.17 (71) with 100 bootstrap replicates under the PROTGAMMALGF model.

**(ii) *Snodgrassella* phylogeny.** *Populibacter corticis* (GCF_001590725.1) was selected as an outgroup for the *Snodgrassella* phylogenomic analysis. The concatenated sequences of 75 *Snodgrassella* and *P. corticis* genomes were extracted from the 254-core-gene supermatrix of *Neisseriaceae* to produce a supermatrix of 76 organisms by 85,654 unambiguously aligned amino acid positions (0.04% missing character states), from which a phylogenomic tree was inferred with RAxML as described above. A leave-one-out analysis was then performed by randomly deleting ≈20% of genes present in our data set. To this end, the corresponding alignments were also reduced to 76 organisms and used to construct 100 data sets of about 70,000 conserved positions by randomly combining alignment files using the script jack-ali-dir.pl from Bio-MUST-Core (available at https://metacpan.org/dist/Bio-MUST-Core). The 100 supermatrices were assembled using SCaFoS v1.30k (70) with default settings. Trees were inferred using RAxML v8.1.17 (71) using the *fast experimental tree search* method and the PROTGAMMALGF model. A consensus tree was built from the set of 100 trees using the program consense v3.695 (from the PHYLIP package [72], but modified to handle long sequence names), with default settings. In order to test the possible influence of a long branch attraction artifact, the outgroup was eliminated from the supermatrix, and a new phylogenomic analysis was performed with the same protocol as described above but on a matrix of 75 organisms by 85,654 unambiguously aligned amino acid positions (0.03% missing character states). We checked the effect of taxon sampling reduction by deleting 40 *Snodgrassella* strains from our supermatrix, while conserving their diversity and the outgroup, using dRep v2.2.3 (43) with default settings. A phylogenetic analysis was inferred with the same protocol described above but using a matrix of 36 organisms by 85,654 unambiguously aligned amino acid positions (0.22% missing character states). Subsequently, nucleotide-based phylogenomic trees were inferred from the 254 core genes. Protein sequence alignments were back-translated by capturing and aligning the corresponding DNA sequences with the program leel (available at https://metacpan.org/dist/Bio-MUST-Apps-FortyTwo). A supermatrix of 76 organisms by 264,981 aligned nucleotides (1.59% of missing character states) was generated using SCaFoS v1.30k (70). A large phylogenomic analysis was inferred using RAxML v8.1.17 (71) with 100 bootstrap replicates under the GTRGAMMA model and using two different partitions for codon positions 1 and 2 together (here, "1&2") and position 3. As for the protein trees, leave-one-out analyses were then performed by random selection of genes to construct

100 data sets of about 100,000 aligned nucleotides. The consensus trees were produced with the same protocol described above, but using the GTRGAMMA model of RAxML v8.1.17 (71). Two leave-one-out analyses were generated: one using only codon positions 1&2 and one using two different partitions for codon positions 1&2 and 3.

**Gut bacterial phylogeny.** A total of 486 genomes representing the main phylotypes of bacteria found in the gut of *Apis* spp. and *Bombus* spp. (excluding *Snodgrassella*) were downloaded from RefSeq, prior to filtering the genomes with CheckM v1.1.3 (63) as described above. Also, 247 additional genomes covering a broad diversity of bacteria and archaea were selected with ToRQuEMaDA (66) using the following options: *dist-threshold of 0.90, kmer-size of 12 and max-round of 10,* and *CheckM activated* (Table S2). To perform a large phylogenomic analysis, we used core genes extracted by CheckM. The CheckM *taxon set* option was first run to produce a lineage set file. Then the CheckM *analyze* option was run, using the bacterial domain set while providing 843 genomes as input (i.e., 75 *Snodgrassella* genomes, 35 *Neisseriaceae* genomes, and the 733 genomes mentioned above). The CheckM *qa* option was then used to produce the marker files. The custom script Checkm-to-OGs.py allowed us to reconstruct OGs from CheckM marker files, selecting only unicopy OGs, thereby producing 76 OGs. The OGs were aligned using MAFFT v7.453 (73), run with the *anysymbol, auto and reorder* parameters. Conserved sites were selected using BMGE v1.12 (69) with moderately severe settings (*entropy cut-off* = 0.5, *gap cut-off* = 0.2). A matrix of 843 organisms by 12,930 unambiguously aligned amino-acid positions (6.72% missing character states) was produced using SCaFoS v1.30k (70), with default settings. A phylogenomic tree was produced with the same protocol as above for large phylogenomics.

**Average nucleotide identity analysis.** Pairwise average nucleotide identities were determined for 75 *Snodgrassella* genomes and the *P. corticis* (GCF_001590725.1) genome using fastANI v1.32 (74) with default settings.

**Specific gene analysis.** A distinct anvi'o pangenomic data set using 75 genomes of *Snodgrassella* and *P. corticis* (GCF_001590725.1) was constructed using the same protocol described above. The custom script anvio_OGs-filtration.py was used to determine which genes occurred specifically in a given group of organisms. This specific gene analysis was performed on a set of 4,685 orthologous groups. We used the same options as for the core genes, with the exception that the number of representatives of a group was set to 0.6 (60%). The inflation number was set to 10 in anvi'o (67) with the value of 60% recovered orthologous groups, even if the inflation parameter was too strict for some of them. All members of a group had to comprise an OG before it was considered specific. An orthologous enrichment was then performed to compensate for the potential lack of sensitivity in orthology inference. In addition, to enrich the OGs with potentially absent members of the group due to the 60% threshold, the enrichment could also add any *Snodgrassella* from other groups to control that the added sequences were indeed specific and not the result of a false orthology inference. We performed the orthologous enrichment with Forty-Two v0210570 (75, 76) (available at https://metacpan.org/dist/Bio-MUST-Apps-FortyTwo). We used the 843 genomes that included the 75 *Snodgrassella* genomes to perform the enrichment. The custom script Confirm-OGs.py was used to validate the specific OGs based on the criteria that all *Snodgrassella* of the considered group, but no *Snodgrassella* from other groups, must be present in the OGs after enrichment. We performed this approach on all groups defined in the present study, as well as all pairwise combinations of them, including one with all groups together (i.e., all S*nodgrassella* strains). All nodes of the tree have also been tested to assess the presence of specific genes shared between groups defined in this study. A total of 107 individual gene phylogenies were computed by aligning the sequences using MAFFT v7.453 (73) with the same options described above, selecting the conserved sites with BMGE v1.12 (69) with the same options described above, and inferring the trees with RAxML v8.1.17 (71) with 100 bootstrap replicates under the PROTGAMMALGF model.

**Metabolic analyses.** The functions of the group-specific OGs were first determined using NCBI COGs (68) if the information was available as a product of the anvi'o workflow. COG pathways and COG functions were used to label the metabolic functions. When COG analyses yielded no results, Mantis (53) was used with default settings, and the information from the consensus annotation file was used to label function. OGs without any hit in COG or Mantis analyses were considered unknown genes, while OGs having hits with proteins of unknown function were classified as "Unknown function."

**Horizontal gene transfer.** GTDBtk (77) was used to infer the GTDB taxonomy (47) of the 561 phylotype genomes (75 *Snodgrassella* and 486 representatives of the other bacterial bee gut phylotypes [described above]). MetaCHIP v1.10.15 (60) with default settings was then used to infer HGT. The group-specific genes were then searched in the MetaCHIP results using BLASTp v2.2.28 (78) with a threshold of 98% of identity on 95% of query length. Twenty-seven out of 107 single-gene phylogenies, containing *Snodgrassella* and at least one additional bacterial genus, generated during the specific gene analysis were manually inspected for HGT.

**Biochemical characterization.** Biochemical characteristics were determined for the bumblebee isolates LMG 28360[T], R-53528, R-54236, LMG 30236[T], and R-53680, and for the type strain of *S. alvi*, NCIMB 14803.

Growth was tested on nutrient agar (Oxoid), tryptic soy agar (TSA) (Oxoid), brain heart infusion (BHI) agar (BD Difco), Columbia agar (Oxoid) supplemented with 5% sheep blood, and All Culture (AC) agar (2% tryptose, 0.3% beef extract, 0.3% yeast extract, 0.3% malt extract, 0.5% dextrose, 0.02% ascorbic acid, 1.5% agar [all wt/vol]) after 2, 3, and 4 days of incubation at 37°C in a $CO_2$-enriched atmosphere (6% $CO_2$, 15% $O_2$) provided by $CO_2$Gen Compact sachets (Oxoid). Hemolysis of sheep blood was checked. Growth was also tested at 37°C in ambient atmosphere (0.04% $CO_2$, 20.95% $O_2$) on AC agar and in an anaerobic atmosphere (9 to 13% $CO_2$, <1% $O_2$) provided by AnaeroGen sachets (Oxoid) on AC agar and on AC agar supplemented with 10 mM $KNO_3$. For the tests mentioned below, cultures were incubated in a $CO_2$-enriched atmosphere. The temperature growth range was tested after 2 days of incubation on AC agar at 4, 15, 20, 28, 37, 40, and 45°C. Cell and colony morphology, motility, oxidase and catalase activities and Gram stain reaction were assessed on cultures grown for 2 days on AC agar at 37°C. Motility was determined by examining wet mounts in broth by phase-contrast microscopy. Oxidase and catalase activities and Gram staining were tested using

conventional procedures (79). The effect of NaCl on growth was investigated in AC broth supplemented with different concentrations of NaCl (0 to 10% with 1% intervals [wt/vol]) after 3 days of incubation. The pH range for growth was evaluated after 3 days of incubation in AC broth buffered at pH 4.0 to 9.0 at intervals of 1 pH unit using the following buffer systems: acetate buffer (pH 4.0 to 5.0), phosphate buffer (pH 6.0 to 8.0), and Tris-HCl (pH 9.0). Nitrate reduction and denitrification and urease, indole and $H_2S$ production were tested using standard microbiological procedures (79) after 72 h of incubation.

Cells were grown on DNase agar (Sigma-Aldrich), AC agar with 0.8% (wt/vol) gelatin (Merck), AC agar with 0.8% (wt/vol) soluble starch, AC agar with 1.3% (w/vol) dried skim milk (Oxoid), and AC agar with 1% (vol/vol) Tween 20 or 80 and were checked for growth after 72 of incubation and for DNase activity, gelatinase activity, and hydrolysis of starch, casein, and Tween 20 and 80, respectively.

**Data availability.** L. Cornet and P. Vandamme have submitted data to the European Nucleotide Archive (ENA) under project accession no. PRJEB47378, read accession no. SAMEA9570070 to SAMEA9570078, and genome accession no. GCA_914768015, GCA_914768025, GCA_914768035, GCA_914768045, GCA_914768055, GCA_914768065, GCA_914768075, GCA_914768085, and GCA_914768095. These database records are publicly available via the ENA website: https://www.ebi.ac.uk/ena/browser/home.

## SUPPLEMENTAL MATERIAL

Supplemental material is available online only.
**TEXT S1**, PDF file, 0.7 MB.
**FIG S1**, PDF file, 0.02 MB.
**FIG S2**, PDF file, 0.1 MB.
**FIG S3**, PDF file, 0.1 MB.
**FIG S4**, PDF file, 0.02 MB.
**FIG S5**, PDF file, 0.03 MB.
**FIG S6**, PDF file, 0.02 MB.
**TABLE S1**, PDF file, 0.4 MB.
**TABLE S2**, PDF file, 0.6 MB.
**TABLE S3**, PDF file, 0.4 MB.

## ACKNOWLEDGMENTS

We thank David Colignon and the CÉCI for help with computing cluster usage and Cindy Snauwaert for performing the phenotypic characterization tests.

This work was supported by a research grant (no. B2/191/P2/BCCM GEN-ERA) financed by the Belgian State—Federal Public Planning Service Science Policy (BELSPO). Computational resources were provided by the Consortium des Équipements de Calcul Intensif (CÉCI) funded by the F.R.S.-FNRS (2.5020.11) and through two research grants to D.B.: B2/191/P2/BCCM GEN-ERA (Belgian Science Policy Office—BELSPO) and CDR J.0008.20 (F.R.S.-FNRS). P.V., N.J.V., D.M., and G.S. were supported through funding from the FWO and F.R.S.-FNRS under the Excellence of Science (EOS) program for the project "Climate Change and Its Impact on Pollination Services" (CLiPS, no. 3094785).

L.C., I.C., D.B., and P.V. conceived the study. I.C. assembled the newly sequenced genomes. R.R.L. carried out the ToRQuEMaDA analyses. L.C. performed the rest of the analyses and drew the figures. L.C., I.C., D.M., D.B., and P.V. wrote the manuscript. P.V., D.B., N.J.V., D.M., and G.S. provided part of the funding. All authors read and revised the manuscript.

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
