## [Reviewer comments · mSystems]

Phylogenomic analyses of *Snodgrassella* isolates from honeybees and bumblebees reveals taxonomic and functional diversity

Luc Cornet, Ilse Cleenwerck, Jessy Praet, Raphaël Leonard³, Nicolas Vereecken, Denis Michez, Guy Smagghe, Denis Baurain, and Peter VANDAMME

Corresponding Author(s): Peter VANDAMME, Ghent University

Review Timeline:

Submission Date:	December 22, 2021
Editorial Decision:	February 19, 2022
Revision Received:	March 29, 2022
Accepted:	April 11, 2022

Editor: Monica Pupo

Reviewer(s): The reviewers have opted to remain anonymous.

Transaction Report:

DOI: <https://doi.org/10.1128/msystems.01500-21>

February 19, 2022

Prof. Peter VANDAMME
Ghent University
Laboratory of Microbiology
K.L. Ledeganckstraat 35
Gent 9000
Belgium

Re: mSystems01500-21 (Phylogenomic analyses of *Snodgrassella* isolates from honeybees and bumblebees reveals taxonomic and functional diversity)

Dear Prof. Peter VANDAMME:

Thank you for submitting your manuscript to mSystems. We have completed our review and I am pleased to inform you that, in principle, we expect to accept it for publication in mSystems. However, acceptance will not be final until you have adequately addressed the reviewer comments.

Please move "Data Summary" paragraph from page 2 to "Data Availability" paragraph at the end of the Materials and Methods section

Preparing Revision Guidelines

Sincerely,

Monica Pupo

Editor, mSystems

Journals Department
Reviewer comments:

Reviewer #1 (Comments for the Author):

In this study, Cornet and colleagues conducted a phylogenetic analysis of the bacterial genus *Snodgrassella*, which is a specialized bee symbiont. They also isolated, characterized, and sequenced several new strains. Overall, the study is rather descriptive, and most of the results were already reported in previous studies. I have multiple concerns with the analyses.

I am not sure why the authors included so many Neisseriaceae genomes. One or two genomes would have been enough to root the tree. The main issue is that it reduces the core genome to only 254 core genes. The inclusion of distant genomes also increases the changes of inferring paralogs as orthologs, which can be problematic since the Neisseriaceae are notoriously affected by HGT and species borders are fuzzy in this family (Hanage et al. 2005). Therefore, it would probably be wiser to only include one of two outgroups and generate a larger core genome. It could also be interesting to rebuild the tree using nucleotide sequence alignments, since this study is focusing on closely related strains.

Specific genes analysis: I am not sure how relevant this analysis is considering that only 254 core genes were included. These analyses are typically conducted on the full core genome or the pan genome of a clade. Considering that there are only 254 highly conserved genes, the authors are likely analyzing genes with housekeeping functions.

The HGT analysis: They only just listed the software (used for metagenomics) and it looks like it was applied to the entire genomes of all bee gut genomes. The paper for that software doesn't seem to mention any type of distinguishing between core and accessory genomes. The authors detected 57 HGTs, and said they found that none of the 107 core genes were present in their HGT results. Then in their conclusion they state the lack of HGT between most the 107 core genes. But again, why are we discussing HGT between core genes unless to prove that they aren't accessory genes, and therefore are reliable for phylogenetic inference?

Assuming gene loss: "In contrast, the 27 specific genes found in other bee endosymbionts or in other bacteria unrelated to the bee gut microbiome suggest that gene loss on other Neisseriaceae was the most plausible evolutionary path of the genes (Figure 4)." I don't think Fig 4 represents this finding or not in an obvious way.

Overall, I found the Methods section difficult to make sense of until after I read the Results & Discussion section several times. The figures are also hard to read and interpret.

Finally, Koch et al. did not really disprove (nor claim to) vertical transmission and co-speciation.

Reviewer #2 (Comments for the Author):

Minor comments

P10, second paragraph: correct leave-on-out to leave-one-out

P12/13/14: Here the term endosymbiont is used for the first time in the manuscript, and in my opinion not correctly.

Endosymbiont usually refers to microbes living intracellularly, or at least extracellularly within the primary body cavity, but not the gut. I suggest changing the expression accordingly, e.g. in gut symbiont, microbiont, or similar...

Figure 3 legend: Clarify that the numbers in the pie charts refer only to unique genes identified in the respective *Snodgrassella* subgroups.

Reviewer #3 (Comments for the Author):

Overall, I found this study to be a very useful, organized, and careful contribution on the diversity of the genus *Snodgrassella*, an interesting host-restricted clade of bacteria in guts of social bees. To date, there has been only a single species name (*S. alvi*), and clearly this does not represent the more complex reality for this 80 million year old symbiont clade. This paper reports phylogenetic work based on whole genomes from 75 isolates, showing that there are probably at least 7 species. This is a big

step towards clarifying the relationships in this group. The choice to formally describe 2 new species is reasonable. It will take a while longer to sort out all of the species in this genus.

The ms did not have page or line numbers, so I am numbering my comments grouped by general topic.

1.

I went over the methods and generally found that this was done very thoroughly and that the approaches are justified. In some cases, alternative approaches were used to verify results, such as with the "leave-one out" phylogenetic analysis. I was unfamiliar with a few of the pipelines, such as Forty-Two, but reading up on these, they seem appropriate. Also results seem quite robust; for example, the position of the Asian *Apis* isolates as a cluster branching at the base of *Snodgrassella*. As mentioned, this was foreshadowed by previous phylogenetic analysis of a single protein coding gene (Powell et al 2015) but this is far more robust, being based on 254 core genes from whole genomes.

2. How these results fit with previously published work

One of the conclusions is that there have been host jumps among *Bombus* species. I think it would be worth mentioning previous results on this, based on possible based on strain-level phylogenetics using single protein-coding gene amplicons (Powell et al. 2015, cited as #65) and also based on experimental transfers for the *Bombus* group5 strains (Kwong et al. 2014, #2).

In particular, Powell et al. 2015 found clades that generally correspond to those in this paper, minus the one from Europe (*Bombus* group4). They found that that one tight *Snodgrassella* clade, corresponding to *Bombus* group5, had a broad host range and was found in hosts in several *Bombus* subgenera. They found that some other clusters appeared host specific, including the cluster in *Bombus nevadensis* and one in *Bombus pensylvanicus* (including *B fervidus* which is close to *B pensylvanicus*). So this agrees with your conclusions that some lineages can jump hosts, even to rather divergent host species, but that some lineages have high host specificity.

Kwong et al. 2014 did experimental transferred *Bombus* group5 (wkB12, wkB29) originating from *Bombus vagans* and *Bombus bimaculatus*, to *Bombus impatiens*, which was robustly colonized, though these strains could not colonize *Apis mellifera*. Nor could *Apis mellifera* wkB1 colonize *Bombus*. So clearly host jumps are possible, but sometimes barred.

Basically, these previous results are highly compatible with your findings and your proposed delineations of species.

3. Sampling and species of *Snodgrassella*

You might emphasize a little more that additional *Snodgrassella* diversity including additional species are almost certain to exist, beyond these 7 in your trees. So far, there are few samples from stingless bees and from *Bombus* from continents other than Europe and North America. On the basis of 16S rRNA gene sequences, Kwong et al. 2017 found distinct lineages in those groups (based on 16S alone). There are ~500 species of stingless bees, and there are many *Bombus* in Asia and some in South America, including some divergent groups, but no genomes or isolates yet for the symbionts.

4. pan genome size in *Snodgrassella*

The specification of the group-specific genes is interesting, mainly because there are so few in *Snodgrassella*. I wonder if any comment can be made on this-is this number unusually low - for *Snodgrassella* overall or for particular groups? Is it similar in *Neisseria* (a related group for which there are lots of genomes)? I am not asking to analyze a lot of genomes in other groups, but perhaps there is information in the literature that makes it possible to comment.

On a related point, it would be interesting to know how many non-core genes are specific below the level of these groups, for example in single strains. Is *Snodgrassella* just low in transferred genes in general? Perhaps you could give the overall size of the pan genome and core genome for each of the hypothesized species, to give an idea of this.

5.

In Fig. S5, how was the distinction between donor and recipient made? (It is stated that the color corresponds to the donor group).

6. Fig 2 presentation

Fig 2 is not well explained. (Possibly some part was cut off in my version?) For example, there is no mention of what the black circles and dots represent. I figured out that they represent host origins, but this isn't stated and the host names are not given. Also, the heat map needs more explanation, and a key. What do the different colors represent? Again, I mostly could figure this out, but you need to actually inform the reader. I think it would be useful to add an A, B, C label for the 3 parts of the figure and to then describe the information in each, in the legend.

7. Fig 3 presentation

Fig. 3 is good overall, as it contains the main results of the paper in one place and is easy to follow. In the pie charts, I wonder if you could make 3 categories (Function unknown, General Function prediction only, Unknown) next to each other, and then use different shades of gray or white for these? I understand that these categories are defined differently, but for the reader they

have similar (lack of) meaning, and this would facilitate more focus on the genes for which there is some functional categorization. As suggested for Fig. 2, labeling the parts A (the tree) and B (the pie charts) would be helpful. Currently, after describing the pie charts, you go back to information on the tree, defining the numbers as boot strap values; this might be confusing to the reader, who is still looking at the numbers on the pie charts.

8. Fig 4 presentation

Fig. 4 would be improved with a little more labeling. On the X axis, perhaps you can write "genes grouped by specificity to species clusters" or similar.

Also (minor) *Bifidobacter* is misspelled.

Parasaccharibacter from bees has been changed to *Bombella* (see

<https://www.microbiologyresearch.org/content/journal/ijsem/10.1099/ijsem.0.004950>) though perhaps your strain has not since I see you also list *Bombella* separately

Lactobacillus-FIRM4 has become *Bombilactobacillus* (you have this update in Fig S4).

9. Tables

Table 3. It would be nice to include a heading at the top: such as strain number.

On the data: it is curious that *S. alvi* wkB2 lacks urease activity when it possesses the full operon for making urease and also possesses the gene for the urea transporter. I wonder if this function is dependent on nitrogen sources in the media?

10. minor typos:

2nd paragraph under Results

"leave-on-out" to "leave-one-out"

Line 9 under Specific gene analysis

S. alvei spelling

EDITOR COMMENT:

February 19, 2022

Prof. Peter VANDAMME
Ghent University
Laboratory of Microbiology
K.L. Ledeganckstraat 35
Gent 9000
Belgium

Re: mSystems01500-21 (Phylogenomic analyses of *Snodgrassella* isolates from honeybees and bumblebees reveals taxonomic and functional diversity)

Dear Prof. Peter VANDAMME:

Thank you for submitting your manuscript to mSystems. We have completed our review and I am pleased to inform you that, in principle, we expect to accept it for publication in mSystems. However, acceptance will not be final until you have adequately addressed the reviewer comments.

Please move "Data Summary" paragraph from page 2 to "Data Availability" paragraph at the end of the Materials and Methods section

Au: this text section was moved.

Line 325-335 of Marked Up Manuscript.

Preparing Revision Guidelines

- Point-by-point responses to the issues raised by the reviewers in a file named "Response to Reviewers," NOT IN YOUR COVER LETTER.
- Upload a compare copy of the manuscript (without figures) as a "Marked-Up Manuscript" file.

- Each figure must be uploaded as a separate file, and any multipanel figures must be assembled into one file.
- Manuscript: A .DOC version of the revised manuscript
- Figures: Editable, high-resolution, individual figure files are required at revision, TIFF or EPS files are preferred

Sincerely,

Monica Pupo

Editor, mSystems

Journals Department
REVIEWER COMMENTS:

Reviewer #1

In this study, Cornet and colleagues conducted a phylogenetic analysis of the bacterial genus *Snodgrassella*, which is a specialized bee symbiont. They also isolated, characterized, and sequenced several new strains. Overall, the study is rather descriptive, and most of the results were already reported in previous studies. I have multiple concerns with the analyses.

Au: We thank the reviewer for the critical comments. We are sorry to read that the reviewer has concerns with our analyses, which is in opposition with the two other reviews. In our revision, we did our best to answer these concerns and clarify the manuscript. We do not agree with the comment concerning the descriptive character of our study, as rigorous phylogenomic analyses accompanied by metabolic modeling of the specific gene content is more than simply 'descriptive'. This paper is the first to report the existence of seven species within the *Snodgrassella* genus, while only one species (*Snodgrassella alvi*) was considered until now. This information provides a novel and very different framework for discussing evolution and species specific adaptations in different hosts. We also report 107 specific genes (along with the scripts to determine them), study their metabolic roles and their origin in the context of lateral gene transfers. Finally, our study characterizes type specimens for two of these new species.

I am not sure why the authors included so many *Neisseriaceae* genomes. One or two genomes would have been enough to root the tree. The main issue is that it reduces the core genome to only 254 core genes. The inclusion of distant genomes also increases the changes of inferring paralogs as orthologs, which can be problematic since the *Neisseriaceae* are notoriously affected by HGT and species borders are fuzzy in this family (Hanage et al. 2005). Therefore, it would probably be wiser to only include one of two outgroups and generate a larger core genome. It could also be interesting to rebuild the tree using nucleotide sequence alignments, since this study is focusing on closely related strains.

Au: The *Neisseriaceae* genomes were added to avoid bias due to HGT in our phylogenomic analyses. The criteria to include a gene in our dataset were very stringent: a gene had to be single copy for all 35 *Neisseriaceae* genomes and have a high (0.8) functional (less than 20% of substitution) and geometric index (less than 20% of gap). With these criteria, the chance to include HGT in the dataset is maximally reduced. In

the revision we nevertheless computed a new large phylogenomic analysis using the orthologous inference realized for the specific gene analysis. The same criteria as above were used for the selection of genes but with only one *Neisseriaceae* genome (*Populibacter corticis* GCF_001590725.1) as outgroup. This resulted in the selection of 916 core genes, from which we inferred a phylogeny with the same protocol as in the original version of our paper (BMGE, SCaFoS, RAxML). This novel phylogenomic tree presented exactly the same topology as the tree computed using 254 core genes, confirming our results. We decided to not include this novel tree in the revised version of our paper since it would complicate the reading of the methods part without providing a useful new result. We nevertheless provided the tree for the reviewer (file labeled 'NOT for publication_protML-916.pdf').

We thank the reviewer for the suggestion of rebuilding the tree in DNA. We used the 254 core genes to build a large DNA phylogenomic analysis and two leave-one-out (as for the protein trees) analyses (using only codon positions 1&2 and using two different partitions for codon positions 1&2 and 3). These DNA sequence-based trees confirmed our results based on protein analysis, with the same delineation of species. We nevertheless included a new supplementary figure with these DNA sequence-based trees because the branching pattern of *Bombus* group2 was not the same in one of these trees. We added this information in the discussion part.

(Line 205-217 & 363 & 389-394 of Marked Up Manuscript).

Specific genes analysis: I am not sure how relevant this analysis is considering that only 254 core genes were included. These analyses are typically conducted on the full core genome or the pan genome of a clade. Considering that there are only 254 highly conserved genes, the authors are likely analyzing genes with housekeeping functions.

Au: The specific genes analysis was not realized on the 254 core genes. We computed a distinct *in vivo* pangenomic dataset using 75 genomes of *Snodgrassella* and *P. corticis* (GCF_001590725.1). The inference of specific genes was made on this whole set of 4685 orthologous groups. The manuscript text was modified to clarify this.

Line 248 & 251-252 of Marked Up Manuscript.

The HGT analysis: They only just listed the software (used for metagenomics) and it looks like it was applied to the entire genomes of all bee gut genomes. The paper for that software doesn't seem to mention any type of distinguishing between core and accessory genomes. The authors detected 57 HGTs, and said they found that none of the 107 core genes were present in their HGT results. Then in their conclusion they state the lack of HGT between most the 107 core genes. But again, why are we discussing HGT between core genes unless to prove that they aren't accessory genes, and therefore are reliable for phylogenetic inference?

Au: This comment is related to the previous one: we did not perform HGT analysis with the set of core genes since the 107 specific genes were not a sub-sample of the 254 but

were inferred from a separate ortholog inference. In other words, we indeed did not try to infer HGTs among core genes for which criteria were set to exclude them!

As species specific genes were analyzed, we wanted to know if these specific genes were vertically or horizontally inherited. We thus used the software MetaCHIP, which indeed can be applied on metagenomic data sets also. MetaCHIP is used to infer HGT among genomes (it is the only accepted input) from members of a microbial community, to identify the potential source of HGT. Adding more genomes will not decrease the efficiency of MetaCHIP, which is why we used it with 561 phylotype genomes in order to maximize the chance to pick up the potential source of HGT. It appeared that none of the 107 genes was affected by HGT according to MetaCHIP. Finally, we did not just 'list the software' but tried to understand these results by a manual examination of single gene phylogeny (27 genes out of 107, present in at least one more bacteria genus than *Snodgrassella*). We concluded that 8 genes might be laterally inherited. The manuscript was modified to be more clear about methods of detection of these 107 genes and HGT inference.

Line 248 & 251-252 of Marked Up Manuscript.

Line 291-293 of Marked Up Manuscript.

Assuming gene loss: "In contrast, the 27 specific genes found in other bee endosymbionts or in other bacteria unrelated to the bee gut microbiome suggest that gene loss on other Neisseriaceae was the most plausible evolutionary path of the genes (Figure 4)." I don't think Fig 4 represents this finding or not in an obvious way.

Au: reference to figure 4 has been removed

Line 524 of Marked Up Manuscript.

Overall, I found the Methods section difficult to make sense of until after I read the Results & Discussion section several times. The figures are also hard to read and interpret.

Au: The methods section has been clarified for the detection of core genes, the specific genes, and HGT inference. We also modified all main figures (Figure 2, 3, 4) with the exception of the graphical abstract. Legends of figures have been modified for clarification.

Line 248 & 251-252 of Marked Up Manuscript.

Line 291-293 of Marked Up Manuscript.

Line 838-876 of Marked Up Manuscript.

Finally, Koch et al. did not really disprove (nor claim to) vertical transmission and co-speciation.

Au: This sentence has been deleted.
Line 419-420 of Marked Up Manuscript

Reviewer #2

Au: We thank the reviewer for the reading and suggestions.

Minor comments

P10, second paragraph: correct leave-on-out to leave-one-out

Au: Done.

Line 363 of Marked Up Manuscript.

P12/13/14: Here the term endosymbiont is used for the first time in the manuscript, and in my opinion not correctly. Endosymbiont usually refers to microbes living intracellularly, or at least extracellularly within the primary body cavity, but not the gut. I suggest changing the expression accordingly, e.g. in gut symbiont, microbiont, or similar...

Au: the use of the term 'endosymbiont' in literature is not consistent (as exemplified in the comments of referee #1 who also uses the 'endosymbiont'), but we agree that it is more appropriate to refer only to intracellular bacteria as endosymbionts, and to refer to *Snodgrassella* bacteria as gut symbionts. We modified the text accordingly.

Figure 3 legend: Clarify that the numbers in the pie charts refer only to unique genes identified in the respective *Snodgrassella* subgroups.

Au: The legend of figure 3 has been modified.

Reviewer #3

Overall, I found this study to be a very useful, organized, and careful contribution on the diversity of the genus *Snodgrassella*, an interesting host-restricted clade of bacteria in guts of social bees. To date, there has been only a single species name (*S. alvi*), and clearly this does not represent the more complex reality for this 80 million year old symbiont clade. This paper reports phylogenetic work based on whole genomes from 75 isolates, showing that there are probably at least 7 species. This is a big step towards clarifying the relationships in this group. The choice to formally describe 2 new species is reasonable. It will take a while longer to sort out all of the species in this genus.

Au: We thank the reviewer for this nice review and suggestions.

The ms did not have page or line numbers, so I am numbering my comments grouped by general topic.

1.

I went over the methods and generally found that this was done very thoroughly and that the approaches are justified. In some cases, alternative approaches were used to verify results, such as with the "leave-one out" phylogenetic analysis. I was unfamiliar with a few of the pipelines, such as Forty-Two, but reading up on these, they seem appropriate. Also results seem quite robust; for example, the position of the Asian *Apis* isolates as a cluster branching at the base of *Snodgrassella*. As mentioned, this was foreshadowed by previous phylogenetic analysis of a single protein coding gene (Powell et al 2015) but this is far more robust, being based on 254 core genes from whole genomes.

Au: We emphasized the use of the 254 core genes now.

Line 371 of Marked Up Manuscript.

2. How these results fit with previously published work

One of the conclusions is that there have been host jumps among *Bombus* species. I think it would be worth mentioning previous results on this, based on possible based on strain-level phylogenetics using single protein-coding gene amplicons (Powell et al. 2015, cited as #65) and also based on experimental transfers for the *Bombus* group5 strains (Kwong et al. 2014, #2).

In particular, Powell et al. 2015 found clades that generally correspond to those in this paper, minus the one from Europe (*Bombus* group4). They found that that one tight *Snodgrassella* clade, corresponding to *Bombus* group5, had a broad host range and was found in hosts in several *Bombus* subgenera. They found that some other clusters appeared host specific, including the cluster in *Bombus nevadensis* and one in *Bombus pensylvanicus* (including *B. fervidus* which is close to *B. pensylvanicus*). So this agrees with your conclusions that some lineages can jump hosts, even to rather divergent host species, but that some lineages have high host specificity.

Kwong et al. 2014 did experimental transferred *Bombus* group5 (wkB12, wkB29) originating from *Bombus vagans* and *Bombus bimaculatus*, to *Bombus impatiens*, which was robustly colonized, though these strains could not colonize *Apis mellifera*. Nor could *Apis mellifera* wkB1 colonize *Bombus*. So clearly host jumps are possible, but sometimes barred.

Basically, these previous results are highly compatible with your findings and your proposed delineations of species.

Au: We thank the reviewer for these suggestions and added several sentences in the manuscript to cite these observations.

Line 424-427 of Marked Up Manuscript.

3. Sampling and species of *Snodgrassella*

You might emphasize a little more that additional *Snodgrassella* diversity including additional species are almost certain to exist, beyond these 7 in your trees. So far, there are few samples from stingless bees and from *Bombus* from continents other than Europe and North America. On the basis of 16S rRNA gene sequences, Kwong et al. 2017 found distinct lineages in those groups (based on 16S alone). There are ~500 species of stingless bees, and there are many *Bombus* in Asia and some in South America, including some divergent groups, but no genomes or isolates yet for the symbionts.

Au: Two sentences have been added to the manuscript to emphasize this point.

Line 406-408 of Marked Up Manuscript.

4. pan genome size in *Snodgrassella*

The specification of the group-specific genes is interesting, mainly because there are so few in *Snodgrassella*. I wonder if any comment can be made on this-is this number unusually low - for *Snodgrassella* overall or for particular groups? Is it similar in *Neisseria* (a related group for which there are lots of genomes)? I am not asking to analyze a lot of genomes in other groups, but perhaps there is information in the literature that makes it possible to comment.

Au: Lu et al., 2019 (<https://www.hindawi.com/journals/ijg/2019/6015730/>) performed a pangenomic analysis of *Neisseria* pathogens. They found 452 and 78 gene families unique to *N. gonorrhoeae* and *N. meningitidis*, respectively. The number of specific genes in *Snodgrassella* thus appears unusually low at first sight. Nevertheless, the determination of the specific genes in the cited study differed from our approach. We first made a selection of potential specific genes among the orthologous groups, which were subsequently verified by orthologous enrichment. If a sequence not belonging to the considered species (*Apis* or *Bombus* groups) was added to the orthologous group during the enrichment, the gene was deleted from the specific gene list. This led to the loss of numerous potential specific genes for each *Apis* or *Bombus* groups. Considering that, we find it difficult to discuss or compare the numbers of specific genes between both studies in our paper. It would be interesting to perform a complete pangenome analysis of the *Neisseriaceae* family using our protocol but that would require an effort that would be more appropriately done in a separate paper.

On a related point, it would be interesting to know how many non-core genes are specific below the level of these groups, for example in single strains. Is *Snodgrassella* just low in transferred genes in general? Perhaps you could give the overall size of the

pan genome and core genome for each of the hypothesized species, to give an idea of this.

Au: We are not sure we understand this comment. Specific genes can be considered as core genes at the hypothesized species level and are provided this way in Figure 3. The size of the *Snodgrassella* core genomes of each of the *Apis* and *Bombus* groups is also provided in Figure 3.

5.

In Fig. S5, how was the distinction between donor and recipient made? (It is stated that the color corresponds to the donor group).

Au: The direction is provided by the software MetaCHIP. One of the first steps of MetaCHIP is to build a reference tree based on 43 universal single-copy genes (SCGs) used by CheckM (similar to what we did for our reference tree, with the exception that we use 76 OGs due to a more precise taxonomic placement). Each gene identified as potential HGT by MetaCHIP has its single gene phylogeny inferred too. These single gene phylogenies are then compared with the reference tree to verify putative HGTs. When a HGT is confirmed, the direction of the transfer is determined by single gene phylogeny. For instance, if a *Snodgrassella* leaf is present in a large *Gilliamella* tree, the donor is determined as *Gilliamella*. This phylogenetic detection of the HGT direction was correctly predicted in 81% in simulations of the MetaCHIP paper. We added the detection mode in the legend of the figure.

Legend has been moved to Figure.

6. Fig 2 presentation

Fig 2 is not well explained. (Possibly some part was cut off in my version?) For example, there is no mention of what the black circles and dots represent. I figured out that they represent host origins, but this isn't stated and the host names are not given. Also, the heat map needs more explanation, and a key. What do the different colors represent? Again, I mostly could figure this out, but you need to actually inform the reader. I think it would be useful to add an A, B, C label for the 3 parts of the figure and to then describe the information in each, in the legend.

Au: we apologize for this. Indeed much information here was missing, including a key for the ANI heatmap, host names, as well as GTDB *Snodgrassella* hits. We extended the margin of the figure to ensure that this information is present in the updated figure. We also added labels (A, B, C) and explained them in the legends.

7. Fig 3 presentation

Fig. 3 is good overall, as it contains the main results of the paper in one place and is easy to follow. In the pie charts, I wonder if you could make 3 categories (Function unknown, General Function prediction only, Unknown) next to each other, and then use different shades of gray or white for these? I understand that these categories are defined differently, but for the reader they have similar (lack of) meaning, and this would

facilitate more focus on the genes for which there is some functional categorization. As suggested for Fig. 2, labeling the parts A (the tree) and B (the pie charts would be helpful. Currently, after describing the pie charts, you go back to information on the tree, defining the numbers as boot strap values; this might be confusing to the reader, who is still looking at the numbers on the pie charts.

Au: Figure 3 is now modified to integrate your suggestion. We have three types of unknown (General Function prediction only, Unknown function, Unseen in databases) with different colors of gray. We also added labels for the two parts and organized the legend accordingly.

8. Fig 4 presentation

Fig. 4 would be improved with a little more labeling. On the X axis, perhaps you can write "genes grouped by specificity to species clusters" or similar.

Also (minor) Bifidobacter is misspelled.

Parasaccharibacter from bees has been changed to Bombella (see

<https://www.microbiologyresearch.org/content/journal/ijsem/10.1099/ijsem.0.004950>)

though perhaps your strain has not since I see you also list Bombella separately

Lactobacillus-FIRM4 has become Bombilactobacillus (you have this update in Fig S4).

Au: Figure 4 has been modified to correct taxonomic names. We also included the suggestion concerning the X axis labeling.

9. Tables

Table 3. It would be nice to include a heading at the top: such as strain number.

On the data: it is curious that *S. alvi* wkb2 lacks urease activity when it possesses the full operon for making urease and also possesses the gene for the urea transporter. I wonder if this function is dependent on nitrogen sources in the media?

Au: strain numbers were provided with the title and legend of the table. We repeated the urease assay for the *S. alvi* type strain and found it to be positive. Table 3 was corrected.

10. minor typos:

2nd paragraph under Results

"leave-on-out" to "leave-one-out"

Au: Corrected.

Line 363 of Marked Up Manuscript.

Line 9 under Specific gene analysis

S. alvei spelling

Au: corrected.

Line 452 of Marked Up Manuscript.

April 11, 2022

Prof. Peter VANDAMME
Ghent University
Laboratory of Microbiology
K.L. Ledeganckstraat 35
Gent 9000
Belgium

Re: mSystems01500-21R1 (Phylogenomic analyses of *Snodgrassella* isolates from honeybees and bumblebees reveals taxonomic and functional diversity)

Dear Prof. Peter VANDAMME:

Your manuscript has been accepted, and I am forwarding it to the ASM Journals Department for publication. For your reference, ASM Journals' address is given below. Before it can be scheduled for publication, your manuscript will be checked by the mSystems production staff to make sure that all elements meet the technical requirements for publication. They will contact you if anything needs to be revised before copyediting and production can begin. Otherwise, you will be notified when your proofs are ready to be viewed.

Publication Fees:

We recognize that the video files can become quite large, and so to avoid quality loss ASM suggests sending the video file via <https://www.wetransfer.com/>. When you have a final version of the video and the still ready to share, please send it to mSystems staff at mssystems@asmusa.org.

For mSystems research articles, if you would like to submit an image for consideration as the Featured Image for an issue, please contact mSystems staff at mssystems@asmusa.org.
